# Imaging More than Skin-Deep: Radiologic and Dermatologic Presentations of Systemic Disorders

**DOI:** 10.3390/diagnostics12082011

**Published:** 2022-08-19

**Authors:** Mehrzad Shafiei, Firoozeh Shomal Zadeh, Bahar Mansoori, Hunter Pyle, Nnenna Agim, Jorge Hinojosa, Arturo Dominguez, Cristina Thomas, Majid Chalian

**Affiliations:** 1Division of Musculoskeletal Imaging and Intervention, Department of Radiology, University of Washington, Seattle, WA 98195, USA; 2Division of Abdominal Imaging, Department of Radiology, University of Washington, Seattle, WA 98195, USA; 3University of Texas Southwestern Medical School, University of Texas Southwestern Medical Center, Dallas, TX 75390, USA; 4Department of Dermatology, University of Texas Southwestern Medical Center, Dallas, TX 75390, USA; 5Department of Internal Medicine, University of Texas Southwestern Medical Center, Dallas, TX 75390, USA

**Keywords:** dermatology, systemic, cutaneous, radiologic features, congenital, genetic, autoimmune, vasculitis, neoplasms, multidisciplinary

## Abstract

Background: Cutaneous manifestations of systemic diseases are diverse and sometimes precede more serious diseases and symptomatology. Similarly, radiologic imaging plays a key role in early diagnosis and determination of the extent of systemic involvement. Simultaneous awareness of skin and imaging manifestations can help the radiologist to narrow down differential diagnosis even if imaging findings are nonspecific. Aims: To improve diagnostic accuracy and patient care, it is important that clinicians and radiologists be familiar with both cutaneous and radiologic features of various systemic disorders. This article reviews cutaneous manifestations and imaging findings of commonly encountered systemic diseases. Conclusions: Familiarity with the most disease-specific skin lesions help the radiologist pinpoint a specific diagnosis and consequently, in preventing unnecessary invasive workups and contributing to improved patient care.

## 1. Introduction

Systemic disorders often present with both cutaneous and radiologic findings. Some presenting skin lesions are disease-specific (e.g., shagreen patch in tuberous sclerosis); others signify internal organ involvement (e.g., erythema nodosum in sarcoidosis, tuberculosis, or inflammatory bowel disease). These cutaneous manifestations sometimes precede more serious findings and should prompt clinicians to investigate further systemic involvement by radiologic examinations. Familiarity with dermatologic signs of these disorders can help radiologists interpret imaging findings more precisely. This may result in early diagnosis of the disease entity and obviate further, often invasive, diagnostic testing, ultimately improving patient management [1,2,3].

This article aims to provide a review of both dermatologic and radiologic manifestations of various systemic diseases, emphasizing findings that are disease-specific. We describe characteristic radiologic findings of common systemic disorders through a multimodality approach after a brief overview of clinical and dermatologic manifestations. Disease entities are broadly categorized into three areas: (1) autoimmune/inflammatory disorders and vasculitides, (2) genetic/congenital disorders, and (3) neoplasms (Table 1). 

## 2. Autoimmune/Inflammatory Disorders and Vasculitides

### 2.1. Dermatomyositis

Dermatomyositis (DM) is a rare autoimmune disease occurring with a bimodal peak incidence of 5–15 at 40–60 years of age with a 2:1 female: male predominance. DM is characterized by skin lesions and myositis [4,5,6]. Skin lesions in DM are often pruritic or burning, photosensitive, and precede myopathic symptoms in 50% of patients [5,7]. Atrophic dermal papules of dermatomyositis (ADPDM, formerly Gottron papules) are pathognomonic violaceous papules and plaques, sometimes with subtle scale, found on the interphalangeal joints of the hands (Figure 1A) [7]. Additional pathognomonic findings include Gottron sign (erythematous macules or patches over the extensor joints) and the heliotrope rash (periorbital erythema with edema, most often affecting the upper eyelids) [7]. Other characteristic skin findings in DM include V sign (erythematous, confluent papules, and plaques over the lower anterior neck and upper chest), shawl sign (violaceous or erythematous papules and plaques over the posterior shoulders and upper back), calcinosis cutis, and nailfold changes (periungual erythema, capillary loop dilation and dropout, and ragged cuticles) [5,6,7].

A large body of evidence exists substantiating an association between dermatomyositis and malignancy, and newly diagnosed patients should be screened for underlying malignancy. Screening typically involves a comprehensive review of the patient’s history, physical examination, and basic labs followed by CT imaging of chest/abdomen/pelvis in cases of high suspicion. Features associated with an increased risk of malignancy include older age at onset (>45), male sex, dysphagia, cutaneous necrosis, cutaneous vasculitis, rapid onset myositis, and elevated inflammatory markers [8,9]. 

Magnetic resonance imaging (MRI) has become a fundamental noninvasive tool for assessment of myositis. MRI provides information to diagnose subclinical disease, estimate disease chronicity, detect optimal site for biopsy, and evaluates treatment response. However, its application is still limited due to cost and availability [4,9,10,11]. Multifocal areas of hyperintensities on fluid-sensitive sequences and areas of enhancement on contrast-enhanced fat-saturated T1-weighted images (T1WI) are seen in active disease. Fatty atrophy (chronic disease) is best visualized on T1-weighted sequences [9,10,11]. Soft-tissue calcifications with varying patterns, including nodular, reticular, amorphous, and sheet-like, occur in up to 70% of patients with DM, predominantly children (Figure 1B). If there is diagnostic uncertainty, plain radiographs are recommended for the initial detection of soft-tissue calcinosis due to their high sensitivity, availability, and low cost (Figure 1C) [4]. 

Thoracic complications occur in >50% of patients and include interstitial lung disease (ILD), aspiration pneumonia, and hypoventilation [6]. While ILD can present as reticulonodular opacities on radiograph, high-resolution computed tomography (HRCT) of the chest is able to differentiate between predominant patterns with higher diagnostic accuracy. These patterns of ILD, which may coexist (Figure 1D) [4] include (A) nonspecific interstitial pneumonia (NSIP, with imaging characteristics of ground-glass opacification (GGO) with reticulation and traction bronchiectasis), (B) organizing pneumonia (OP, seen as peripheral bilateral consolidation and patchy GGOs) and to a lesser extent (C) usual interstitial pneumonia (UIP, manifesting as peripheral reticulation, traction bronchiectasis, and honeycombing), all with basilar predilection. Another complication of DM is the involvement of pharyngeal muscles, which may result in dysphagia, and aspiration pneumonia. This could be measured by an esophagogram, which is a dynamic fluoroscopic swallow study [4,6].

### 2.2. Sarcoidosis

Sarcoidosis is a granulomatous disease with unknown etiology and variable prevalence, with a predilection for African American women in their third to fifth decades of life [12,13,14,15]. Patients may be asymptomatic or experience a wide spectrum of multiorgan involvement [13,14] demanding radiologic investigation for diagnosis and follow-up [12,13,15]. Twenty to thirty-five percent of patients with sarcoidosis develop skin lesions, which often manifest at the onset of systemic illness and, thus, can provide diagnostic clues. Specific skin lesions are those with epithelioid granulomas without associated inflammation on histopathology (e.g., lupus pernio, Darier–Roussy). Erythema nodosum (EN, reported in up to 25% of cases) consists of tender self-limiting erythematous subcutaneous nodules most commonly present on the shin. Lupus pernio (LP) describes characteristic chronic violaceous indurated papulonodules with central face distribution. It is strongly associated with extracutaneous involvement, specifically pulmonary disease [12,14,15,16]. Sarcoid may also present as firm, well-demarcated, skin-colored to violaceous papules with a predilection for the face (Figure 2A). Of note, sarcoidosis has been reported to mimic other diseases, including herpes zoster, chronic cutaneous lupus erythematosus, ichthyosis, and psoriasis [17].

Thoracic involvement is seen in nearly 90% of patients with chronic sarcoidosis, and usually presents with symmetric, bilateral, hilar, and right paratracheal adenopathy with amorphous or cloud-like calcified lymph nodes (Figure 2B). Parenchymal involvement and pulmonary embolism are other chest manifestations of sarcoidosis. Parenchymal involvement usually reveals bilateral, symmetric, small, rounded opacities with apical predominance on chest radiography. Irregular (2–5 mm) nodules with perilymphatic predilection causing irregular micronodular thickening of fissures and interlobular septa can be seen on HRCT. Parenchymal involvement can progress to irreversible disease manifesting as mid- to upper lung reticular opacities radiating from the hila (Figure 2C) [13,14,18,19,20]. 

Bone involvement mostly affects phalanges and toes with a pathognomonic lacy lytic appearance (Figure 2D), or less commonly, purely lytic lesions. MRI is more sensitive when evaluating appendicular skeleton, axial skeleton, and marrow involvement. MRI can show variable-sized with T1 hypointensities and T2 hyperintensities (Figure 2E). Chronic or healed lesions may present with signal intensities consistent with fat or fibrosis [13,14,15,18]. Cardiac sarcoidosis accounts for up to 85% of sarcoidosis-related deaths in Japan. Cardiac involvement patterns on late-gadolinium-enhanced cardiac MRI is nonspecific. However, it is mostly seen as patchy and multifocal late gadolinium enhancement in the basal segments of the septum and the lateral wall. On Fluorodeoxyglucose–positron emission tomography (FDG-PET), active inflammation is seen as 18F-FDG uptake with or without a perfusion defect [14].

Neurosarcoidosis (NS) can occur in any part of the brain. Cranial neuropathy is the most common CNS presentation, and the optic nerve is the most commonly involved nerve, which may be seen on MRI as thickening of the nerve with abnormal enhancement [13,14,18]. Intraparenchymal findings most commonly present as multiple small T2-hyperintense and T1-hypointense foci within the periventricular white matter. Plaque-like or nodular thickening of the hypothalamus and pituitary gland with T1 isointensity, T2 hypointensity, and marked enhancement may also present on MRI. Leptomeningeal and dural involvement are seen, respectively, as enhancing nodular (Figure 2F) and plaque-like thickening on postcontrast T1-weighted fat-saturated sequences. Dural nodularities also manifest as foci of hypointensity on T2-weighted images (T2WI). In early disease, spinal cord involvement presents as nodular leptomeningeal enhancement along the spinal cord. Late manifestations of cord involvement include elongated eccentric intramedullary T1 hypointensity and T2 hyperintensity with patchy post-contrast enhancement, typically in the cervicothoracic spine, with subsequent cord atrophy [13,14,21]. 

### 2.3. Scleroderma

Scleroderma, or systemic sclerosis, is an autoimmune fibrosing disorder, consisting of two subsets: diffuse cutaneous systemic sclerosis (dSSc; a systemic disease characterized by widespread involvement of any organ system with a prevalence of 20/100,000 and peak incidence in females between 30 and 50 years old), and limited cutaneous systemic sclerosis (lcSSc; a disease characterized by manifestations of the CREST syndrome (calcinosis cutis, Raynaud phenomenon, esophageal dysmotility, sclerodactyly (Figure 3A), and telangiectasia). Cases without skin changes but other systemic manifestations have been reported as well and are described as systemic sclerosis sine scleroderma [22,23,24]. The skin is the main organ involved in scleroderma, and disease subsets are differentiated by the degree of skin involvement [23]. Raynaud’s phenomenon, cutaneous sclerosis, nailfold and fingernail alterations, cutaneous ulcerations, telangiectasias, “salt and pepper” hyper/hypopigmentation (Figure 3B), and calcinosis cutis are common skin manifestations seen in scleroderma patients [4,23]. Cutaneous sclerosis begins in the fingers, extends proximally to the metacarpophalangeal joints, and affects the face at an early stage. The skin becomes pale and hairless as the skin folds disappear. The current literature supports using high-frequency ultrasound for quantitative and reliable evaluation of dermal thickness in patients with SSc. Dermal thickness has been shown to be inversely corelated with blood perfusion. Additionally, US elastography has been shown to be of value in the evaluation of the skin in SSc [23,25,26,27,28,29]. Claw hands (Figure 3A), thin lips, sharp nose, and a characteristic “mouse face” appearance can result from this [23]. 

Morphea, previously called localized cutaneous sclerosis (LSc), is a distinct disorder from systemic sclerosis and causes limited sclerosis of the skin (Figure 3C) with rare subcutaneous tissue and bony involvement. Patients with morphea do not typically have diffuse skin involvement or systemic manifestations. Due to confusion with SSc and to decrease patient anxiety, use of “localized scleroderma” is discouraged [22].

Soft-tissue and musculoskeletal manifestations include hand edema, acro-osteolysis (involving palmar surface with progression to pencil-tip appearance), calcinosis (Figure 3D,E), flexion contractures, and arthralgias [4,23]. Bone resorption may also be seen at the ribs, distal radius and ulna, distal clavicle, and mandible [23]. The GI tract can be involved from the esophagus to anus in patients with SSc. Up to 90% of SSc patients have significant esophageal motility abnormalities. Esophagogram shows a patulous and dilated esophagus, with no peristalsis, and the presence of gastroesophageal reflux [4,23]. Esophageal dilation and shortening below the level of the aortic arch can be seen on the classic chest radiograph. Dilated tubular esophagus without peristalsis with gastro-esophageal junction widening and contrast medium regurgitation back to the esophagus may be seen on barium swallow [29]. 

Pulmonary involvement is the leading cause of death in SSc and most commonly includes SSc-related ILD (SSc-ILD) and pulmonary hypertension. SSc-ILD is characterized by conventional radiography showing faint bibasilar reticulation to thick peripheral interstitial opacification accompanied by traction bronchiectasis and volume loss [23]. HRCT has a higher sensitivity for detecting SSc-ILD, including mostly NSIP and UIP [6,23]. Reticulation with a predilection for posterior basilar aspects of the lower lobes is seen in both UIP and NSIP (Figure 3F). GGOs and microcytic honeycombing are characteristic of NSIP and UIP, respectively; however, they can be found in both [4,6,23]. 

### 2.4. Celiac Disease

Celiac disease (CD) is an autoimmune disorder reaching a 1% incidence in most populations caused by gluten sensitivity. It is associated with intestinal and extraintestinal manifestations, including osteoporosis, iron deficiency, and skin lesions [30,31]. The most commonly associated skin manifestations are dermatitis herpetiformis (DH) and psoriasis. DH, seen in more than 85% of patients, is a chronic relapsing vesiculobullous skin disease. It is characterized by symmetric, pruritic, erythematous papules, and vesicles with a predilection for the extensor surfaces of the extremities, scalp, and buttocks (Figure 4A). Psoriasis is also associated with CD and is characterized by well-demarcated erythematous, silver-scaled plaques on extensor surfaces [30].

Intestinal involvement is the leading cause of morbidity, mainly secondary to malabsorption as well as an increased risk for mucosa-associated lymphoid tissue (MALT) lymphoma [25]. Small bowel follow-through under fluoroscopy shows jejunal dilation, fold thickening, decreased jejunal fold with increased ileal fold (so-called “reversal of fold pattern”), hypomotility, and transient intussusceptions (Figure 4B) [32,33]. CT and MRI are more sensitive and can better delineate bowel wall thickening, mesenteric lymphadenopathy, duodenojejunal fatty proliferation, hypervascular mesentery, and hyposplenism. CT enterography (CTE) with intravenous and oral contrast can show ulcers, strictures, mucosal enhancement, increased splanchnic circulation, dilated vasa recta, reversed jejunoileal fold pattern, and ileal fold thickening (Figure 4C) [32,33,34]. MR enterography (MRE) has shown comparable sensitivity to CTE for the diagnosis of intestinal inflammation and has been used for the diagnosis of CD and its complications, particularly malignancy [31,33]. 

### 2.5. Granulomatosis with Polyangiitis

Granulomatosis with polyangiitis (GPA, formerly Wegener’s granulomatosis) is a rare necrotizing granulomatous vasculitis, which is associated with positive cytoplasmic antineutrophil cytoplasmic antibodies (c-ANCAs). GPA primarily affects small vessels in the upper and lower respiratory tracts and kidneys, and medium-sized arteries. The peak incidence is from 46 to 60 years of age, with equal prevalence in males and females [34,35,36]. Skin involvement is seen in about 50% of patients and is polymorphous, including palpable purpura (cutaneous small vessel vasculitis is most common) (Figure 5A), nodules, vesicles, and necrotic lower extremity ulcers on a background of livedo reticularis. Pyoderma-gangrenosum-like ulcerations are a less common finding (Figure 5B) [34]. 

GPA affects the lungs in about 50–90% of patients [35]. Chest radiograph is able to detect large pulmonary nodules. HRCT shows more detailed pathologies, including variable-sized nodules (±cavitation) and GGOs (Figure 5C). Smooth or nodular thickening of the tracheobronchial tree, occasionally multifocal, causes luminal narrowing. The tracheal posterior membrane and subglottic region are most commonly involved [35,36]. Nearly all patients with GPA have ear, nose, and throat involvement at early stages of the disease [35]. Sinonasal involvement manifests as mucosal thickening, bony erosions, and neo-osteogenesis, which together are specific for GPA. Mucosal nodular thickening most commonly involves the maxillary sinuses and is mostly detected on MRI. Erosion and punctuate areas of bone destruction primarily involve the anterior ethmoidal region and is best visualized by CT [35,36]. Saddle nose deformity and perforation of the nasal septum can be present (Figure 5D) [29]. In 6% of patients, contiguous spread of inflammation can lead to the involvement of the skull base, resulting in cranial neuropathy, and is seen as cranial nerve enhancement and thickening on MRI [35,36]. 

### 2.6. Polyarteritis Nodosa

Polyarteritis nodosa (PAN) is a rare systemic ANCA-negative vasculitis involving small-to-medium-sized muscular-walled arteries with peak incidence in the fifth to sixth decades of life. The kidneys, skin, peripheral nerves, and gastrointestinal tract are most involved. Tissue biopsy along with clinical and laboratory data is diagnostic for PAN [36,37,38,39]. In one-third of patients, cutaneous manifestations are the primary feature of the disease. Common skin manifestations include palpable purpura, livedo reticularis, and nodules (Figure 6A). In addition, some patients might only manifest cutaneous lesions without systemic involvement, termed cutaneous PAN. The most frequent of these manifestations is the presence of nodules on the lower legs, which are often found in different stages of development. Less common skin features include urticaria, superficial phlebitis, distal necrosis, and splinter hemorrhages [39].

Imaging can assist with the early diagnosis of PAN. Catheter angiography, CT angiography (CTA), and MR angiography (MRA) can be used to evaluate disease burden, evaluate cases where tissue biopsy is inconclusive or limited, and assess mesenteric (Figure 6B) or renal circulation (Figure 6C). Possible findings include multiple aneurysms (1–5 mm) and irregular constrictions occurring at small- and medium-sized arterial bifurcations. Unlike conventional angiography, CTA and MRA are less invasive and capable of evaluating arterial wall thickening and end-organ damage [36,37,38].

### 2.7. Behcet’s Disease

Behcet’s Disease (BD) is a systemic vasculitis of unknown etiology involving different-sized vessels presenting between the second and fourth decades of life with a higher prevalence around the historical Silk Road. Since various organs may be involved, use of appropriate imaging modalities is mandatory for the assessment of disease extent [40,41]. BD is marked by recurring oropharyngeal ulcers, genital ulcers, and ocular involvement. Earlier onset of mucocutaneous manifestations in BD indicates a worse prognosis [42]. Oral and genital involvement manifests as recurrent painful vesiculopustules evolving into apthous ulcers (Figure 7A,B). Oral ulcerations are often large and appear in groups, with frequent recurrence. Genital ulcers are smaller and occur less frequently. Other common cutaneous lesions include erythema-nodosum-like nodules, pseudofolliculitis, papulopustular lesions, acneiform nodules, and superficial thrombophlebitis (Figure 7C,D) [41,42].

Vascular involvement in BD includes venous and arterial occlusion and aneurysmal dilation involving the abdominal aorta and pulmonary arteries. Early-stage findings include irregular wall thickening, perivascular fat stranding, and delayed mural enhancement, whereas late-stage features of arterial vasculitis include stenosis and aneurysmal formation on CT and MRI. Multiple, bilateral pulmonary artery aneurysms are a rare but characteristic feature of BD (Figure 7E), which may present as parahilar nodular opacities on chest radiograph. CTA/MRA delineate vessels and collaterals, the presence of thrombus, and evidence of mediastinal involvement [40,43]. CNS disease manifests in 10–50% of patients. Acute attacks initially involve the basal ganglia or brainstem, with extension to the diencephalic structures, and show contrast enhancement and scattered areas of T2 hyperintensity (Figure 7F). Months later, small scattered hyperintense lesions present in the periventricular white matter [40,43].

## 3. Genetic/Congenital Disorders

### 3.1. Tuberous Sclerosis Complex

Tuberous sclerosis complex (TSC) is a hamartomatous disease due to a mutation of TSC1/ TSC2 genes with a prevalence between 6.8 and 12.4/100,000. It commonly involves the CNS, heart (rhabdomyomas in 50–65% of patients), kidneys, and lungs [2,15,44,45,46,47]. A number of skin lesions are diagnostic for TS, including facial angiofibromas (malar hamartomatous red nodules) (Figure 8A), hypopigmented macules (“ash leaf spots” and “confetti” lesions), shagreen patches (grayish-green/ light brown lesions in the lumbosacral region), and periungual fibromas (“Koenen tumors”, soft periungual nodules) [2,15,45,46].

CNS involvement mostly includes tubers, white matter radial migration lines (RMLs), subependymal nodules (SENs), and subependymal giant cell astrocytomas (SEGAs). Cerebral tubers are commonly multiple and bilateral with frontal lobe predilection. They appear from infancy to adulthood, ranging from T1 hyperintensity to T1 hypointensity and from T2 hypointensity to T2 hyperintensity. Calcified lesions are considered to have T2 hypointensity. On CT, tubers are hypodense [2,46,47]. RMLs, extending outward from the ventricular surface toward the cortex, appear as curvilinear or straight T2/FLAIR hyperintensities on MRI (Figure 8B). SENs are hamartomas usually scattered along the ependymal surface of the lateral ventricles with a predilection for the foramen of Monro; SENs are considered to have T1 hyperintensity and T2 isointensity (hypointensity if calcified). Head CT in children detects more than 80% of calcified SENs. SEGAs are low-grade vascular tumors that are frequently bilateral and located near the foramen of Monro, with the potential to cause severe hydrocephalus. SEGAs are iso- or hyperdense on CT with frequent calcifications and T1 hypo/isointensity and T2 hyperintensity on MR imaging with intense enhancement [2,46,47].

Renal involvement includes angiomyolipomas (AMLs), which are composed of varying amounts of blood vessels, smooth muscle, and fat, and occur in 55–75% of TSC patients, and are commonly multiple and bilateral. On ultrasonography, AMLs are homogeneous or heterogeneous hyperechoic lesions. Propagation velocity artifact is diagnostic for AMLs. CT and MRI are more specific for diagnosis by detection of macroscopic fat. CT scan can detect areas with fat attenuation. On MRI, bulk fat is seen as T1 and T2 hyperintensity, corresponding to T1 and T2 fat-suppressed hypointensity. Peripheral linear signal loss, termed India ink artifact, is seen on T1-weighted opposed-phase images (Figure 8C) [46,47].

Thoracic manifestations include lymphangioleiomyomatosis (LAM; 1–3%). LAM almost exclusively occurs in women, and is characterized by diffuse, thin-walled, well-circumscribed lung cysts of varying sizes and uniform distribution in bilateral lungs (Figure 8D). They are associated with recurrent pneumothoraxes, chylothoraces, and enlarged lymph nodes [2,46,47]. Skeletal manifestations of TS include focal or diffuse, irregular cyst-like lesions, with peripheral sclerosis (found on conventional radiographs or HRCT) and periosteal new bone formation occurring in the short tubular bones, spine, pelvis, and calvaria (hyperostosis of the inner table) [15,47].

### 3.2. Neurofibromatosis Type 1

Neurofibromatosis type 1 (NF1, von Recklinghausen’s disease) results from a mutation of the neurofibromin gene (incidence 1:2500–1:3000). Manifestations generally occur during the first decade of life [15,48,49,50]. Hallmark cutaneous manifestations of NF1 include café-au-lait spots (well-defined and homogenous brown macules or patches), freckling (axillary or inguinal) (Figure 9A), Lisch nodules (benign iris hamartomas), and neurofibromas [15,49,50]. Cutaneous neurofibromas are benign tumors that develop during childhood, can occur with pruritis, and range in number from several to thousands of lesions (Figure 9B) [15,50].

Malignancies, including gliomas and peripheral nerve sheath tumors (PNFs), occur 4–6 times more commonly in NF-1 patients than in the general population. MRI is commonly used for the evaluation of neoplastic lesions [50]. Optic gliomas (pilocytic astrocytoma) are the most common tumor in NF-1 (18%) and can occur bilaterally [15,43,44]. On MRI, gliomas are seen as T1-hypointense and T2-hyperintense with variable contrast enhancement (Figure 9C). Gliomas in NF-1 have also been reported in the cortex, cerebellum, and basal ganglia [50].

Neurofibroma, the most common benign tumor in NF-1, is a peripheral nerve sheath tumor and includes cutaneous, subcutaneous, spinal, and plexiform neurofibromas [15,51]. Spinal neurofibromas, usually originating from cervical peripheral nerves, show T1 hypointensity and T2 hyperintensity with intense contrast enhancement and may contain central necrosis (Figure 9D) [44]. Plexiform neurofibromas grow along nerves longitudinally, may undergo malignant transformation, and can cause hyperostosis of nearby osseous structures [15,50]. On MRI, masses show T1 hypointensity and T2 hyperintensity with variable enhancement and central hypointensity (target sign) (Figure 9E) [49,50]. Some patients with NF-1 develop malignant peripheral nerve sheath tumors that mostly arise from plexiform neurofibromas. Characteristic findings include a large size, lack of a target sign, irregular shape, and unclear margins seen as T1 hyperintensity with inhomogeneous or poor enhancement on MRI [15,49,50]. Structural brain changes include macrocephaly, increased white matter, enlargement of the corpus callosum, cerebral asymmetries, and unidentified bright objects (presenting as T2 hyperintensity without mass effect) [50].

### 3.3. Sturge–Weber Syndrome

Sturge–Weber syndrome (SWS; encephalotrigeminal angiomatosis) is a sporadic disease characterized by unilateral facial capillary malformation (port-wine stain), eye involvement, and brain abnormalities [47,51,52]. Port-wine stains are blanching dermal venular malformations (pink-to-bright-red patches and plaques) that classically present unilaterally in the first branch of the trigeminal nerve (forehead, eyelids, temple), but occasionally may involve the neck, chest, trunk, and limbs (Figure 10A) [2,51,52].

In patients with SWS and leptomeningeal venular malformations, characteristic radiologic findings include decreased perfusion of the cortex and white matter with progressive severe ipsilateral parieto-occipital cortical hemiatrophy, calcification, lateral ventricle choroid plexus enlargement, calvarial thickening, and sinus hyperpneumatization [2,46,51,52]. MRI shows extent of the pial angioma [46,51]. Contrast-enhanced T2-weighted FLAIR images improve detection of leptomeningeal disease compared to post-contrast T1-weighted images (Figure 10B,C). Bone marrow signal changes are also observed in the skull or facial bones in the majority of young patients [46]. Calcification is better visualized on CT (Figure 10D) and susceptibility-weighted MR imaging (Figure 10E). Tram-track or railroad-track calcification of the adjacent sulci is also visible on skull radiographs [51,52].

### 3.4. PHACES Syndrome

PHACES syndrome is a neurocutaneous vascular disorder [53] observed in 2–3% of infantile hemangioma cases with a 9:1 female predilection [52,54]. It is described as posterior fossa malformation, infantile hemangioma (IH), arterial anomalies, cardiac defects, eye, and sternal abnormalities, which rarely all coexist simultaneously [52,53,54]. Facial hemangiomas are the hallmark of PHACES syndrome. IHs can be absent or present at birth but often are visible by the end of infancy. IHs associated with PHACE syndrome are large (>5 cm) or segmental. Hemangiomas in PHACE syndrome usually occur on the face, affecting regions of facial developmental prominences, especially the cephalic segment [53,54].

Extracutaneous manifestations of PHACES syndrome mostly include structural brain and cerebrovascular anomalies, leading to severe morbidity [46]. Structural brain anomalies include hypoplasia or agenesis of the posterior fossa (Figure 11A,B), cerebrum, corpus callosum, or septum pellucidum [52,53,54]. Cerebrovascular anomalies include hypoplasia, absence, or an abnormal course of major cerebral vessels, persistent embryonic arteries, and aneurysms. Arterial anomalies are much more common than venular anomalies [53,54]. MRI and MRA of the brain and neck are diagnostic modalities if PHACE syndrome is suspected [53,54].

### 3.5. Nevoid BCC Syndrome (NBCCS)

Nevoid BCC syndrome is an autosomal dominant (AD) disorder also known as basal cell nevus syndrome or Gorlin–Goltz. NBCCS arises from the mutation of PTCH1, PTCH2, or SUFU genes, causing overstimulation of the sonic hedgehog signaling pathway and development of benign and malignant neoplasms [2,15,55]. Patients with NBCCS present primarily in the third decade of life with skin-colored or pigmented dome-shaped basal cell carcinomas resembling benign nevi, with a predilection for sun-exposed areas. Asymmetric palmar and plantar pits (shallow depressions in stratum corneum) are early diagnostic clues present in approximately 85% of patients [2,15].

Keratocystic odontogenic tumors (75% of patients), are aggressive cystic lesions located more frequently in the mandible (Figure 12A) [15]. Other skeletal features include rib abnormalities (bifid, fused, hypoplastic, or splayed) mostly involving the third to fifth ribs (Figure 12B), thoracocervical vertebral fusion, occipitovertebral junction malformations, flame-shaped phalangeal lytic bone lesions, and shortened fourth and fifth metacarpals (Figure 12C) [2,15,49]. Desmoplastic medulloblastoma is an important associated tumor typically positioned laterally in the cerebellar hemispheres. Other imaging findings include early calcification of the falx cerebri (Figure 12D) and osseous bridging of the sella turcica, which are best seen on CT scans [2,15].

### 3.6. Hereditary Hemorrhagic Telangiectasia

Hereditary hemorrhagic telangiectasia (HHT), also known as Osler–Weber–Rendu syndrome, is an AD disorder. HHT is characterized by recurrent epistaxis, mucocutaneous and visceral telangiectasias, and arteriovenous malformations (AVMs) [56,57]. Epistaxis, caused by nasal mucosa telangiectasia, is the most common manifestation of HHT and is often apparent by age 10. Skin involvement appears by the age of 40 as multiple telangiectasias of the lips, tongue, face, trunk, arms, and fingers (Figure 13A–C) [56,58].

The arteriovenous malformations in HHT are direct connections between the pulmonary artery and vein through a thin-walled aneurysm without any capillary vessels. They often present as well-defined homogeneous opacities and lobulated enlarged arteries and veins on chest radiographs [50,51]. CT shows ground-glass nodules, with solid components and the architecture of the feeding artery and vein (Figure 13D,E). An enhanced phase with thin slice thicknesses is also acquired from the upper abdomen to evaluate the presence of hepatic vascular fistulas (Figure 13F) [58].

### 3.7. Birt–Hogg–Dube’ Syndrome

Birt–Hogg–Dube´ syndrome is an AD disorder caused by mutations in the folliculin tumor suppressor gene. BHD presents with cutaneous lesions, renal tumors, and lung cysts that may lead to spontaneous pneumothorax [59,60,61,62]. Cutaneous findings of BHD include fibrofolliculomas and trichodiscomas (benign hamartomas of hair follicles) and acrochordons (skin tags). These findings classically appear in the third and fourth decade of life, often affecting the face, neck, and trunk. Fibrofolliculomas typically appear as 2–4 mm skin and white, smooth, dome-shaped papules, or as comedonal or cystic variants [59,60].

Lung cysts on chest CT are characterized by multiple well-defined, thin-walled cysts of various shapes and sizes (<1 cm). These cysts most commonly occur bilaterally with lower and medial lobe predominance. Subpleural (Figure 14A) and fissural cysts, those involving the costophrenic sulci, as well as cysts abutting pulmonary veins or arteries, are helpful in diagnosis. The surrounding lung parenchyma is usually normal [61,62]. The most ominous complication of BHD syndrome is renal cancer occurring in renal cysts (Figure 14B), which should be evaluated by CT or MRI [62].

### 3.8. McCune–Albright Syndrome

McCune–Albright syndrome (MAS) is a rare congenital disorder caused by a mutation in the GNAS1 gene with a mosaic pattern that primarily affects females. MAS is characterized by the clinical triad of fibrous dysplasia (FD; monostotic or polyostotic), skin abnormalities (classical cafe´-au-lait skin pigmentation), and hyperfunctioning endocrinopathies (most commonly precocious puberty) [15,63,64]. MAS presents primarily with skin changes. Shortly after birth, classic cafe´-au-lait macules or patches manifest with jagged irregular borders that have been compared to the coastline of Maine. These lesions often occur in the lumbosacral area and buttocks and tend to be unilateral (ipsilateral to skeletal lesions) [15,64].

Fibrous dysplasia typically presents during childhood and in 50% of cases manifests in the craniofacial bones, pelvis, femur, and tibia. FD extends from the marrow to the cortex as irregular enlargement with polycyclic or multiloculated appearance. Classic imaging findings include medullary ground-glass lytic areas (Figure 15A,B) with thin cortices and endosteal scalloping. The pattern may vary from predominantly sclerotic to cystic. A characteristic sign of FD is the “shepherd’s crook” deformity of the femur, which is caused by multiple cortical microfractures [15,63,64]. MRI findings of FD include intermediate-to-low signal intensity on T1WI and intermediate-to-high signal intensity on T2WI and STIR images [63,64].

### 3.9. Fong Disease

Fong disease, also known as nail–patella syndrome (NPS), is an AD condition affecting mesodermal and ectodermal tissue. NPS is caused by mutations in the LIM homeodomain transcription factor, which results in developmental defects of the glomerular basement membrane (nephropathy), dorsoventral limb structures, nails, and the anterior segment of the eyes [65,66,67,68,69]. Nail dysplasia (triangular nail lunulae) and patellar aplasia or hypoplasia are diagnostic features of NPS. Nail dysplasia usually presents at birth as anonychia, hemianonychia, longitudinal ridging and splitting, and spoon-shaped flaky nails. The thumbs typically show the most severe symptoms [66,69].

Characteristic quartet radiographs findings of “NPS knee” include: patellar aplasia or hypoplasia, anterior surface flattening of the medial femoral condyle, and a short lateral femoral condyle with anterior surface prominence on lateral radiograph (Figure 16A,B). Genu valgus leading to early symptomatic knee arthritis may also occur [66,69]. Luxation of the radial head (Figure 16C) and (“Frog’s prongs”) posterior iliac horns (Figure 16D) are other common imaging findings of NPS [66,67,68,69].

### 3.10. Maffucci Syndrome

Maffucci syndrome (MS) is a congenital nonhereditary disorder of early mesodermal dysplasia, typically appearing before puberty. MS is characterized by multiple enchondromatosis (Ollier disease), venous malformations (hemangiomas and lymphangioma), and malignant transformation [15,70]. Cutaneous lesions of MS are dark blue or skin-colored patches, or nontender nodules, which arise from capillary or cavernous hemangiomas of the subcutaneous tissues. These lesions primarily affect distal extremities (especially hands and feet) and can be associated with phleboliths, which are typical calcifications of the vessels [15,70]. The association between phleboliths and multiple enchondromas on hand radiographs is characteristic of Maffucci syndrome [71].

Enchondromas are benign-appearing radiolucent lesions, occurring asymmetrically on the metacarpals and phalanges of the hands (Figure 17). Radiographs demonstrate expansile remodeling of the adjacent bone with cortical thinning and endosteal scalloping. Tumors outside of the phalanges (commonly involving long bones of the arms and legs) show chondroid matrix mineralization with ring-and-arc calcifications. These lesions may present as a diffusely punctuated or stippled pattern or have a light trabeculation appearance [15,71,72]. The major complication of enchondromatosis is chondrosarcoma. Imaging findings of chondrosarcoma include deep or extensive endosteal scalloping with cortical erosion and periosteal reaction with an enhancing soft-tissue component that is best seen on MRI [71,72].

### 3.11. Buschke–Ollendorff Syndrome

Buschke–Ollendorff syndrome (BOS) is a rare, often benign, autosomal dominant skin disorder. BOS is characterized by connective tissue nevi and osteopoikilosis (OPK) sclerotic bony lesions. BOS is caused by a mutation of the LEMD3 gene, affecting bone morphogenesis, and the TGF-b gene, affecting skin elastin formation [15,73,74,75]. Skin lesions typically appear on the extremities, trunk, lower back, and buttocks within the first year of life. OPK often occurs after puberty at the end of long bones or phalanges, and also in tarsal and carpal spongiosa bones [15,76]. Cutaneous findings of BOS include dermatofibrosis lenticularis disseminate, characterized by symmetrical yellow or skin-colored small papules, or more frequently, larger yellowish nodules with an asymmetrically grouped distribution (Figure 18A) [15,73,75].

OPK are asymptomatic dense “bony islands” (Figure 18B) presenting as numerous well-defined symmetric densities. These lesions are found incidentally on radiographs as sclerotic densities and give the bone a mottled appearance. In some cases, OPK lesions resemble osteoblastic metastases. However, normal bone scintigraphy in OPK excludes other differential diagnoses [15,73,74,75]. Melorheostosis (Figure 18C), another BOS association, is a dense, irregular, eccentric hyperostosis of the cortex with a distinct demarcation border that causes irregular thickening of cortical bone with a melting wax appearance on imaging [73,76,77].

### 3.12. Peutz–Jeghers Syndrome (PJS)

Peutz–Jeghers syndrome (PJS) is a rare AD disease caused by a mutation in the STK11 tumor suppressor gene. PJS is characterized by benign gastrointestinal hamartomatous polyps, mucocutaneous pigmentation, and a high tendency for malignant transformation [78,79]. It is associated with an increased risk of GI malignancies (colon, pancreas, small bowel, gastroesophageal, and stomach) and extraintestinal malignancies (breast, gynecologic, pancreatic, lung, and testis (Figure 19A)) [80,81,82,83].

Mucocutaneous pigmented macules often precede GI symptoms, occur in infancy, and are found predominantly around the mouth, nostrils, fingers, toes, and both dorsal and volar aspects of hands and feet. Macules are dark brown or blue-brown (Figure 19B), 1–5 mm in size, and found in 95% of patients with PJS. While cutaneous lesions tend to fade after puberty, oral buccal mucosal pigmentations are usually persistent [81,82,83]. Clinically, PJS polyps cause colicky abdominal pain and lower GI bleeding. Radiographically, polyps are pedunculated with a typical lobulated pattern at enteroscopy or colonoscopy, presenting as iso- to hyperechoic intraluminal masses on CT and causing multiple nodular filling defects on fluoroscopic study. MR enterography demonstrates T2-isointense intraluminal masses, which are homogeneously enhanced after contrast administration (Figure 19C) [82,83].

## 4. Neoplasms

### 4.1. Melanotic Melanoma

Melanoma is a highly invasive cutaneous cancer that is well-known for early metastasis arising from small primary tumors with a 5-year survival rate of 27%. Primary melanomas often arise in sunlight-exposed areas within a pre-existing melanocytic nevus or occur de novo and are divided into pigmented (melanotic) and nonpigmented (amelanotic) subtypes. They may represent one or more of the following ABCDE features: asymmetry, an irregular border, color variegation, a diameter greater than 6mm, and evolving morphology [84,85,86].

Metastatic melanomas most commonly present near the surgical site of previous lesions as macules, papules, or nodules, or may manifest as firm palpable nodules that may be subcutaneous (Figure 20A). Additionally, melanomas may present as angiomatoid metastasis (soft-tissue mass with hemorrhagic and necrotic components) (Figure 20B) or hematoma-like metastases with ecchymosis [87]. Melanoma often spreads to regional lymph nodes (Figure 20C) via lymphatics but can spread hematogenously to the liver, lung, and brain [85,87]. Brain metastases are best depicted on post-contrast MRI; as melanin reduces T1WI relaxation time, lesions with an adequate amount of melanin represent high SI on T1WI (Figure 20D), which is an uncommon finding in other cancers.

Fluorodeoxyglucose–positron emission tomography (FDG-PET) is another modality commonly used in detecting metastatic foci, revealing high FDG uptake due to increased metabolic activity [1,84,85,86,87]. A novel machine-learning-based model known as radiomics allows for the translation of the medical images to quantitative data. Radiomic features of 18F-FDG-PET have been investigated in recent studies to predict prognosis in patient with metastatic melanoma before immunotherapy treatment [1,84,85,86,87,88].

### 4.2. Kaposi Sarcoma

Kaposi sarcoma (KS) is a low-grade endothelial neoplasm associated with human herpesvirus 8 (HHV-8) infection. This vascular tumor affects blood vessels and lymphatic channels, manifesting as one of four clinical subtypes: classic KS (sporadic or Mediterranean), endemic KS (African), iatrogenic KS (immunosuppression-related), and epidemic KS (AIDS–related). KS is currently the most prevalent AIDS-related malignancy. KS skin involvement is polymorphous, ranging from violaceous macules and papules (Figure 21A) to exophytic tumors with associated lymphedema. AIDS-related KS also frequently affects the upper body, head, and neck [88,89,90,91,92,93]. Twenty-two percent of patients manifest oral cavity lesions as the presenting sign. Oral KS presents simultaneously with cutaneous and visceral involvement in up to 71% of patients with HIV. Lesions are polymorphic and most frequently affect the hard palate, gingiva, and dorsal tongue [92].

While classic and endemic KS is often restricted to skin manifestations, iatrogenic KS and AIDS-related KS frequently involve visceral organs. The most notable sites of involvement include the GI tract, lymph nodes, lung, and liver (Figure 21B). A characteristic imaging finding of AIDS-related KS is prominent enhancement after contrast injection. Enlarged enhancing lymph nodes are found in 80% of patients with disseminated KS. Hepatosplenomegaly and periportal hyperechoic nodules with associated enhancement on delayed scans are seen on CT and MRI (Figure 21C). Chest CT shows symmetrical ill-defined nodules (Figure 21D) with peribronchovascular distribution (flame-shaped lesions) and surrounding ground-glass opacities (halo sign) [90,91,93].

## 5. Conclusions

Various systemic conditions have specific or nonspecific dermatologic and imaging features. Simultaneous consideration of imaging findings and dermatologic manifestations helps in more precise imaging interpretation and narrows down the differential diagnosis toward the final diagnosis. Sometimes the cutaneous manifestation of a systemic disease is predictive of systemic involvement, e.g., pulmonary hypertension and ILD in lcSSc vs. dSSc, respectively. Familiarity with the most disease-specific skin lesions help the radiologist pinpoint a specific diagnosis and, consequently, in preventing unnecessary invasive workups and contributing to improved patient care.

## Figures and Tables

**Figure 1 diagnostics-12-02011-f001:**
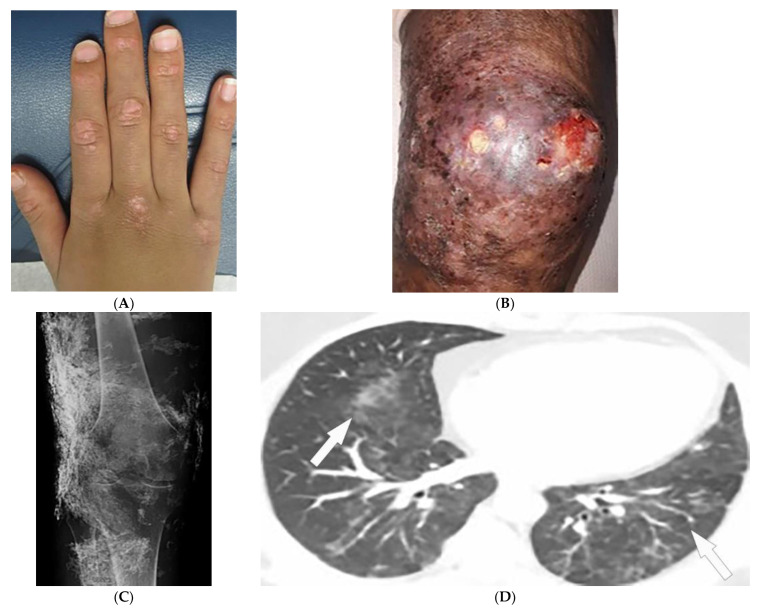
Dermatologic and radiologic images representative of dermatomyositis: (**A**) Flat-topped papules “Gottron’s papules” over the dorsum of the hand of a 19-year-old female. (**B**) Skin erythema and ulceration of the knee due to soft-tissue calcifications in a 45-year-old female. (**C**) AP radiograph of the same knee displays sheet-like soft-tissue calcifications. (**D**) Axial chest CT image (lung window) demonstrates patchy bilateral ground-glass opacities (GGOs) indicative of dermatomyositis-associated interstitial lung disease (ILD) (arrows).

**Figure 2 diagnostics-12-02011-f002:**
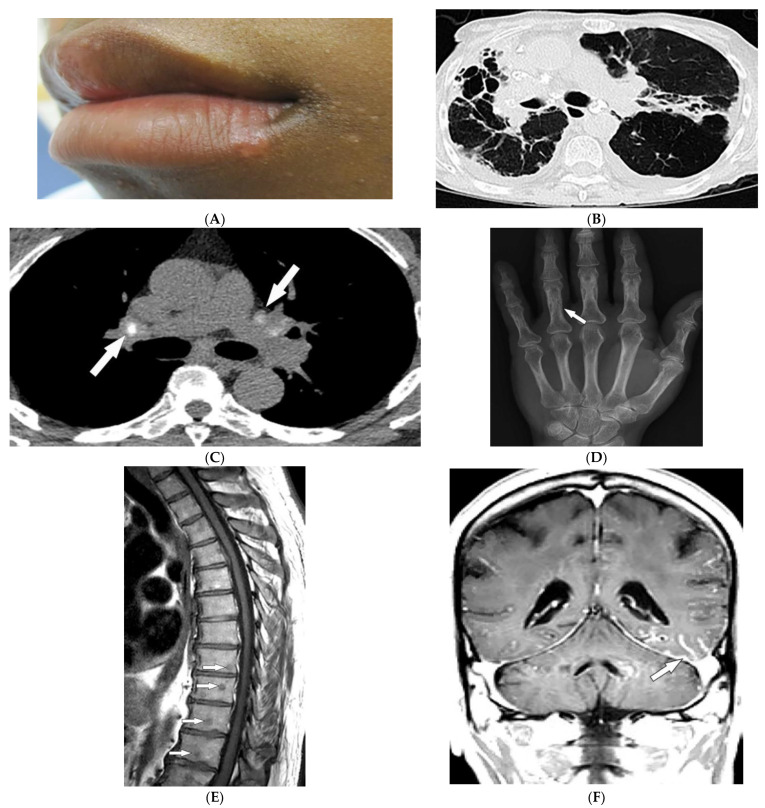
Dermatologic and radiologic images characteristic of sarcoidosis: (**A**) Skin-colored papules on mucosal and cutaneous lips seen in a 19-year-old male with skin sarcoidosis. (**B**) Axial chest HRCT image (lung window) of a 56-year-old male demonstrates multiple areas of bronchiectasis, cysts, and architectural distortion consistent with end-stage pulmonary sarcoidosis. (**C**) Axial chest CT image (soft-tissue window) in a 35-year-old male showing calcified hilar lymph nodes (arrows). (**D**) AP radiograph of the hand of a 45-year-old male demonstrates lacy lytic osseous sarcoid of multiple phalanges (arrow). (**E**) Sagittal T1-weighted image (T1WI) of the thoracic spine in a 40-year-old woman shows multiple well-circumscribed sarcoid marrow lesions (arrows). (**F**) Coronal contrast-enhanced T1WI of the brain of a 42-year-old male shows leptomeningeal enhancement (arrow).

**Figure 3 diagnostics-12-02011-f003:**
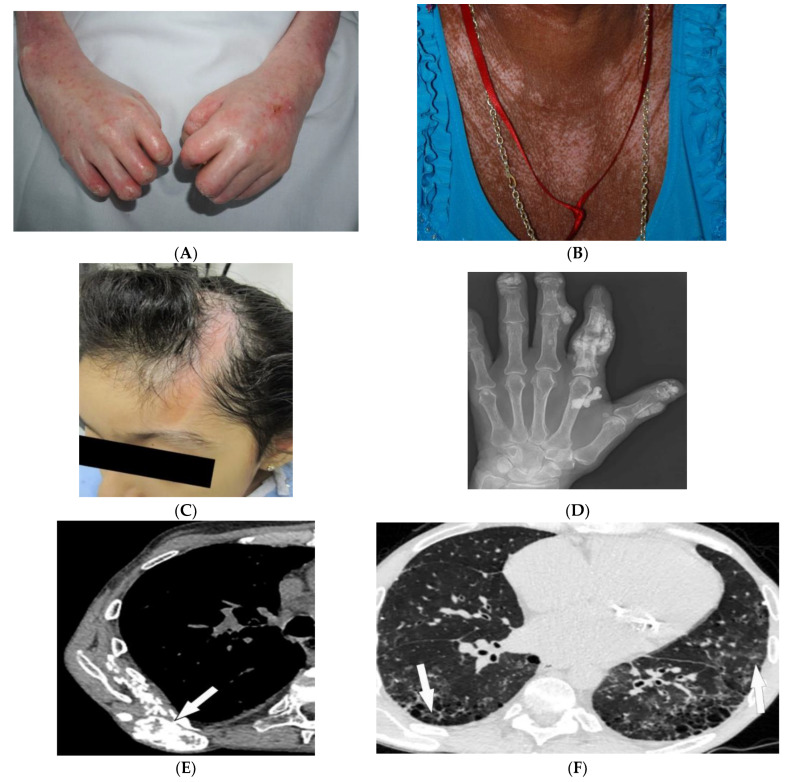
Dermatologic and radiologic images representative of scleroderma: (**A**) Taut shiny skin affecting the arms and hands with associated contractures of the fingers of a 56-year-old female patient with SSc. (**B**) “Salt and pepper” hyper/hypopigmentation of SSc in a 50-year-old female on the chest. (**C**) Indurated bound-down atrophic linear plaque with alopecia on the forehead and scalp of a 19-year-old female consistent with linear morphea. (**D**) AP radiograph of the hand of a 42-year-old female demonstrates soft-tissue calcifications and acro-osteolysis of the scleroderma. (**E**) Axial chest HRCT image (soft-tissue window) in a 42-year-old male shows the right back musculature calcinosis (arrow). (**F**) Axial chest HRCT image (lung window) in 38-year-old male demonstrates fine reticulonodular opacities (arrow) consistent with scleroderma-associated nonspecific interstitial pneumonia (NSIP).

**Figure 4 diagnostics-12-02011-f004:**
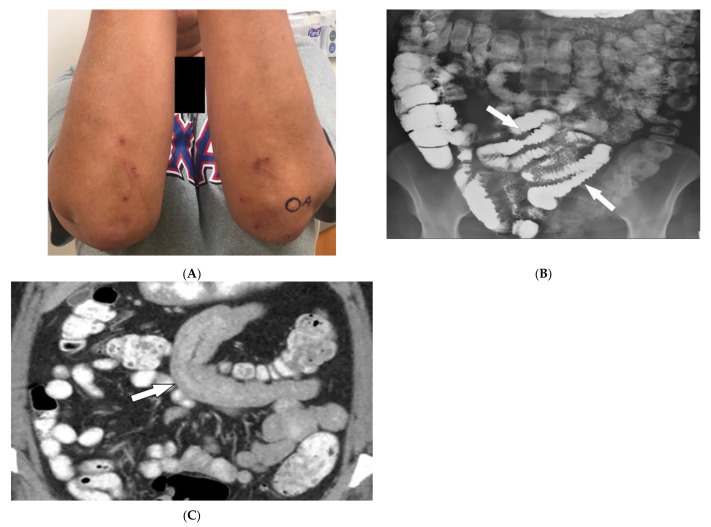
Dermatologic and radiologic images illustrative of Celiac disease: (**A**) Clustered vesicles over the bilateral extensor elbows of a 25-year-old female compatible with dermatitis herpetiformis. (**B**) Fluoroscopic small-bowel follow-through in a 22-year-old female demonstrates reversal of jejunal and ileal folds with more prominent folds in the ileum (arrow). (**C**) CT enterography coronal image (soft-tissue window) in a 42-year-old female shows thickening of the small-bowel folds (arrow).

**Figure 5 diagnostics-12-02011-f005:**
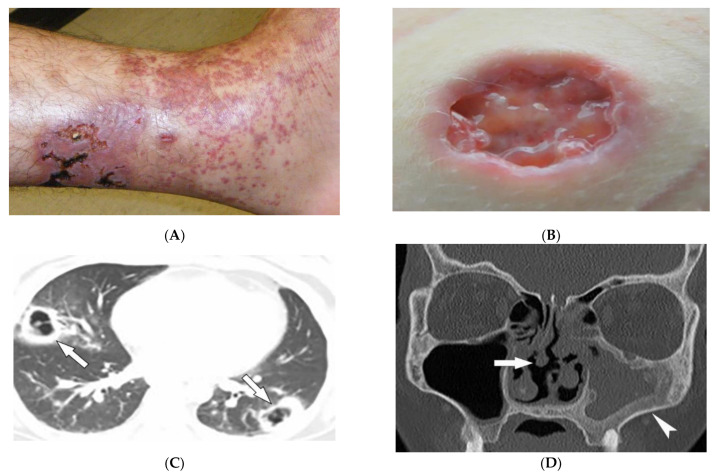
Dermatologic and radiologic images representative of polyangiitis: (**A**) Palpable cutaneous purpura with retiform eschars and ulceration in a 42-year-old male. (**B**) Ulcer with jagged undermined borders in a 26-year-old female with granulomatosis with polyangiitis resembling pyoderma gangrenosum. (**C**) Axial chest HRCT image (lung window) of a 36-year-old male demonstrates bilateral cavitary lung lesions with a ground-glass halo, suggesting surrounding hemorrhage (arrows). (**D**) Coronal maxillofacial CT (bone window) of a 41-year-old female shows sequelae of chronic sinusitis secondary to granulomatous inflammation, including septal perforation (arrow) and left maxillary sinus hyperostosis (arrowheads).

**Figure 6 diagnostics-12-02011-f006:**
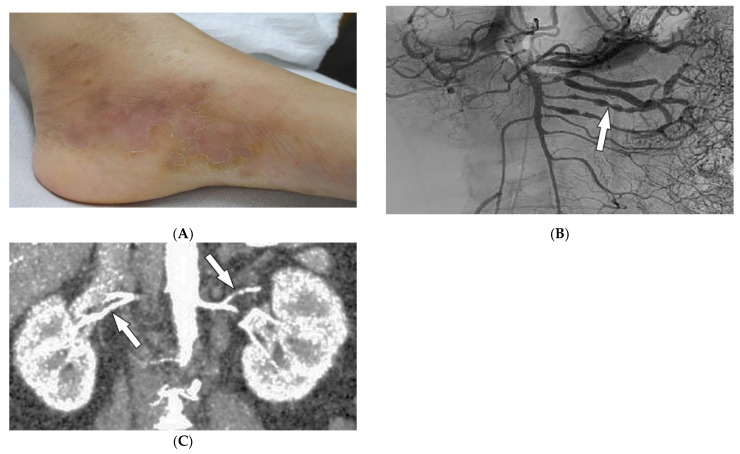
Dermatologic and radiologic images representative of polyarteritis nodosa: (**A**) Painful clustered subcutaneous nodules and plaques on the foot of a 36-year-old female. (**B**) Mesenteric angiogram shows the beaded appearance of multiple mesenteric arteries (arrow). (**C**) Coronal CT angiogram of the abdomen showed a beaded appearance of bilateral renal arteries (arrows).

**Figure 7 diagnostics-12-02011-f007:**
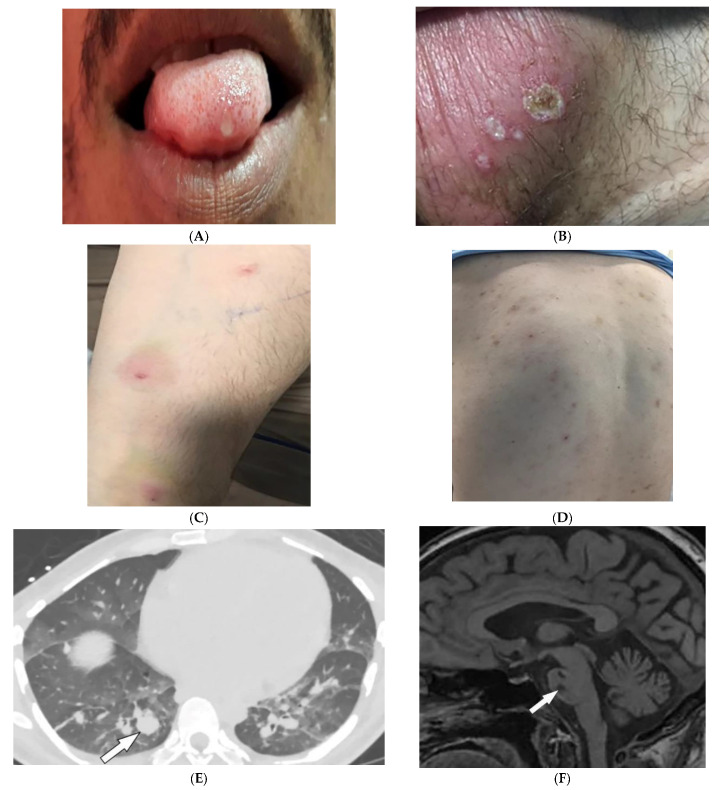
Dermatologic and radiologic images illustrative of Behcet’s disease. (**A**) Ulceration on the tongue of a 52-year-old male. (**B**) Scrotal ulcerations and erosions (**C**) Image of the forearm of a 35-year-old male showing a positive pathergy test. (**D**) Acneiform eruption with residual post-inflammatory hyperpigmentation. (**E**) Axial HRCT of the chest with contrast in the lung window shows multiple bilateral pulmonary artery aneurysms (arrow). (**F**) Sagittal FLAIR MRI of the brain displays pontine involvement in neuro-Behçet’s disease (arrow).

**Figure 8 diagnostics-12-02011-f008:**
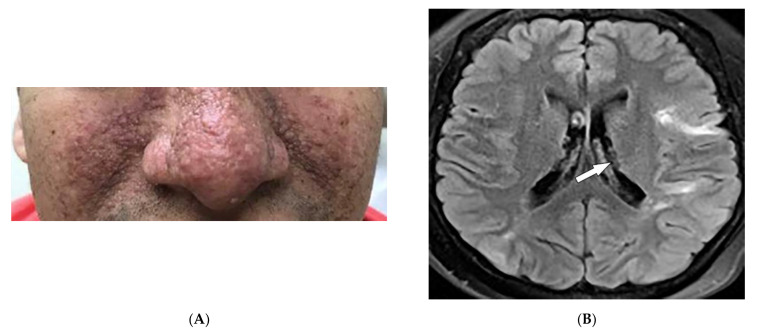
Dermatologic and radiologic images characteristic of tuberous sclerosis (TS): (**A**) Confluent small angiomatous (erythematous, glistening) papules on the cheek and nose of a 44-year-old man consistent with neurofibromas of TS. These lesions were not present during the first few years of life. (**B**) Axial FLAIR MRI of the brain shows linear hyperintensity extending radially from the left subcortical white matter to the gray–white junction, representing subependymal tubers (arrow). (**C**) Coronal T1WI of the abdomen demonstrating a large fat-containing right renal mass, representing angiomyolipoma (arrow). (**D**) Coronal HRCT of the chest demonstrates innumerable thin-walled cysts in a diffuse distribution and a right pneumothorax. Findings are consistent with lymphangioleiomyomatosis (LAM) with spontaneous pneumothorax.

**Figure 9 diagnostics-12-02011-f009:**
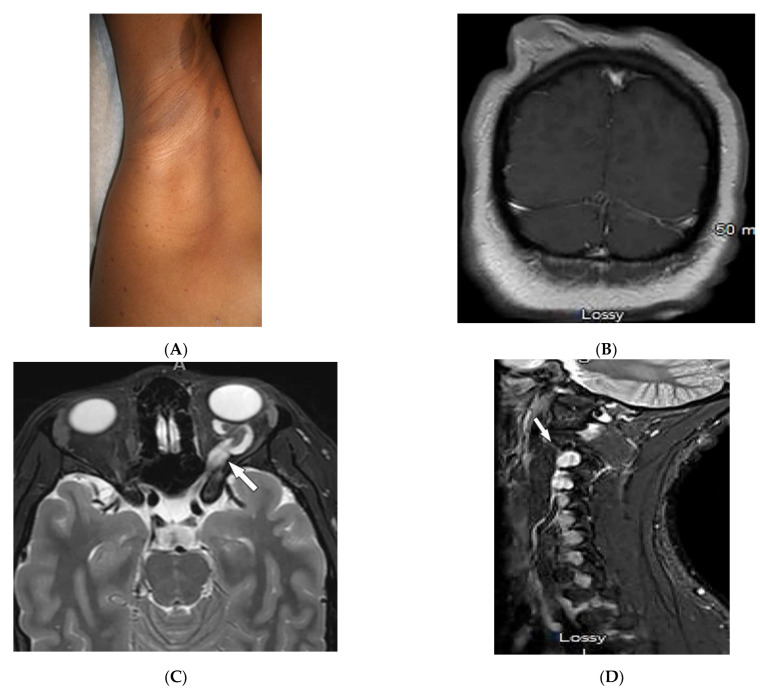
Dermatologic and radiologic images representative of neurofibromatosis type 1 (NF1): (**A**) Café-au-lait macules on the upper arm and multiple small macules on the axillae (axillary “freckling”). (**B**) Coronal post-contrast T1WI of the brain demonstrates diffuse cutaneous neurofibromatosis. (**C**) Axial T2WI of the brain shows diffuse thickening of the left optic nerve (arrow), consistent with optic glioma. (**D**) Sagittal T2WI of cervical spine demonstrates intraneural foraminal neurofibromas (arrow). (**E**) Sagittal T1WI of cervical spine in a 56-year-old male demonstrates plexiform neurofibroma (arrow).

**Figure 10 diagnostics-12-02011-f010:**
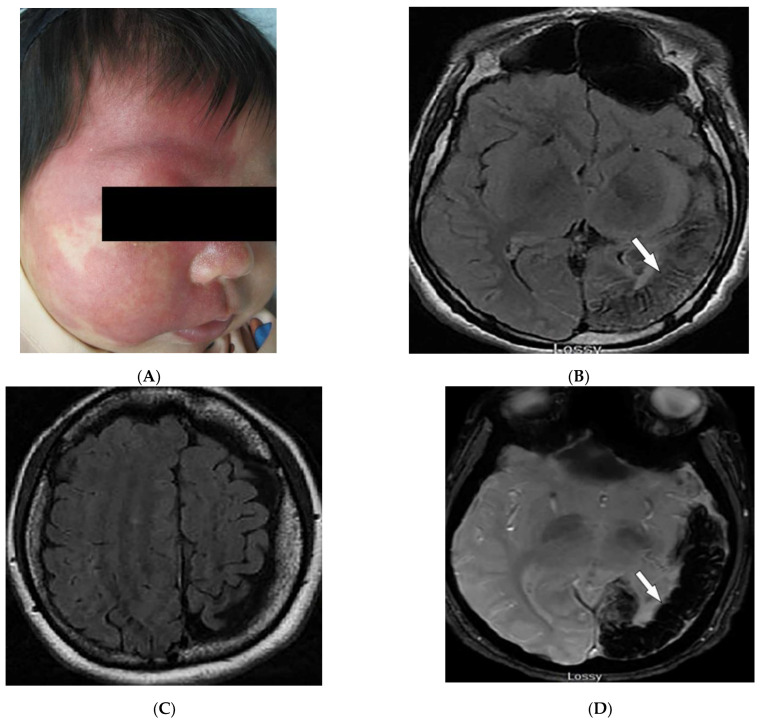
Dermatologic and radiologic images representative of Sturge–Weber syndrome: (**A**) Sharply marginated port-wine stain involving V1-V2 distribution. (**B**) Axial FLAIR MRI of the brain shows atrophy of the left parietal and occipital lobes (arrow). (**C**) Axial FLAIR MRI of the brain depicts diffuse atrophy of the left cerebral hemisphere. (**D**) Axial susceptibility-weighted MR image of the brain demonstrates corresponding loss of signal in the left parietal and occipital lobes (arrow), likely secondary to microcalcifications. (**E**) Sagittal CT of the brain demonstrates tram-track calcifications of the parasagittal parieto-occipital lobe (arrow) in a 21-year-old male.

**Figure 11 diagnostics-12-02011-f011:**
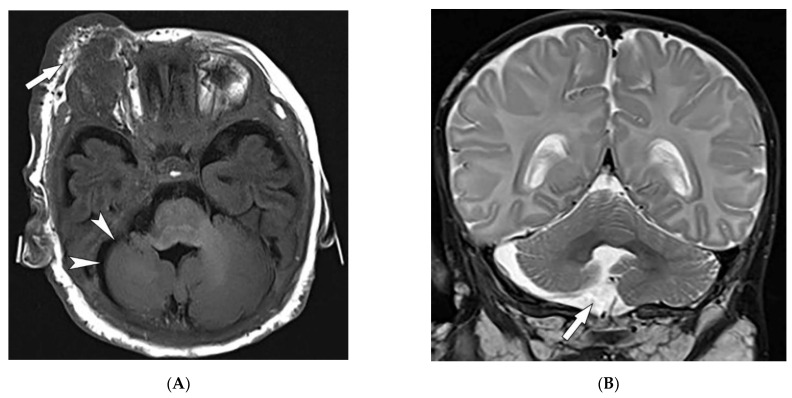
Radiologic images illustrative of PHACES syndrome: (**A**) Axial T1WI of the brain in a 19-year-old male shows right periorbital cutaneous and deep subcutaneous hemangioma (arrow) and ipsilateral right cerebellar hemisphere hypoplasia (arrowheads). (**B**) Coronal T2WI of the brain on the same patient demonstrates right cerebellar hypoplasia (arrow).

**Figure 12 diagnostics-12-02011-f012:**
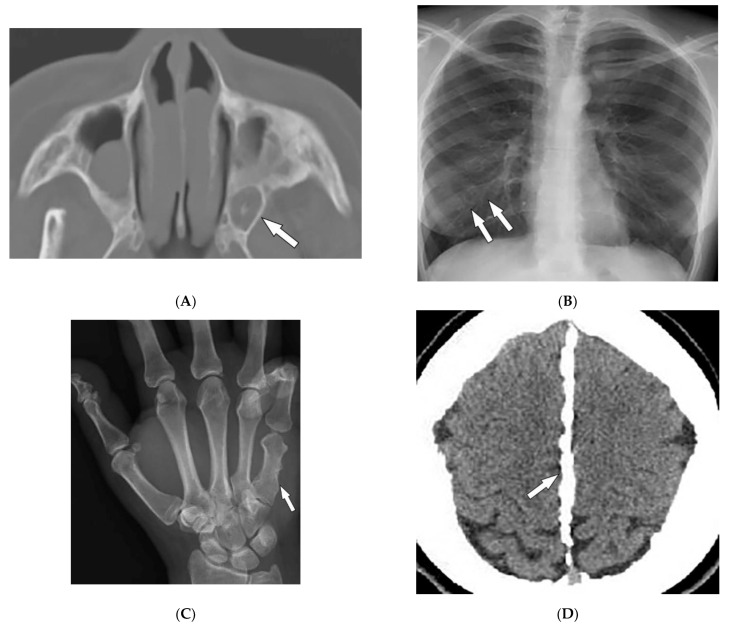
Radiologic images illustrative nevoid basal cell carcinoma syndrome: (**A**) Axial CT of the maxillofacial in bone window in a 24-year-old male demonstrates a left odontogenic keratocyst (arrow). (**B**) Frontal chest radiograph shows multiple bifid ribs (arrows). (**C**) AP radiograph of the hand presents a shortened fifth metacarpal bone (arrow). (**D**) Axial CT of the head in soft-tissue window demonstrates falx calcifications (arrow).

**Figure 13 diagnostics-12-02011-f013:**
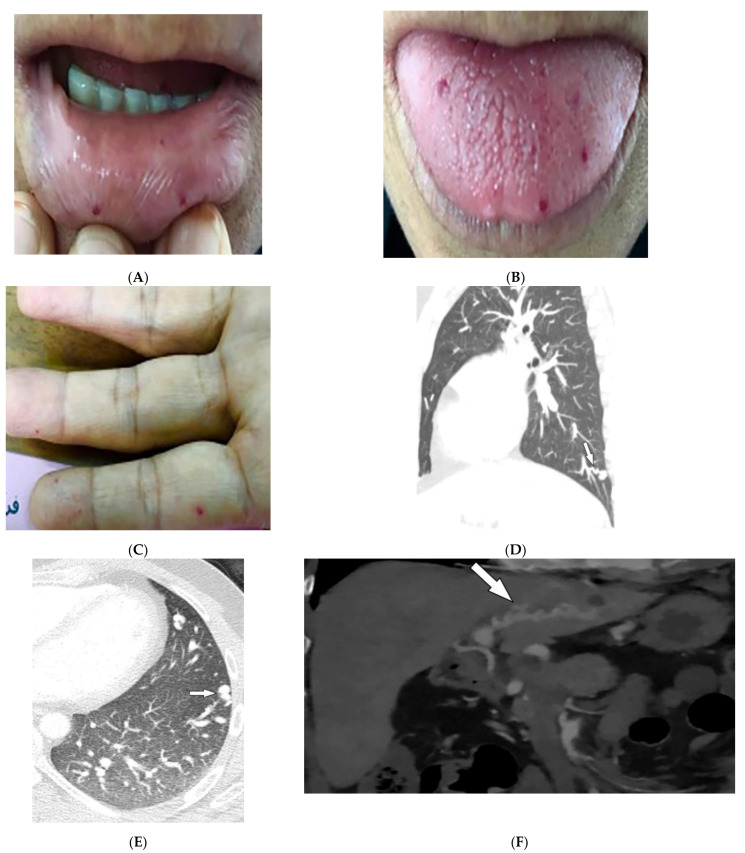
Dermatologic and radiologic images illustrative of hereditary hemorrhagic telangiectasia: (**A**,**B**) Mucosal telangiectasia in a 38-year-old male. (**C**) Telangiectasias over the fingers. (**D**) Sagittal and (**E**) axial contrast-enhanced chest CT of a 17-year-old male demonstrates pulmonary AVM (arrows). (**F**) Coronal contrast-enhanced CT of the abdomen shows hepatoportal AVM (arrow).

**Figure 14 diagnostics-12-02011-f014:**
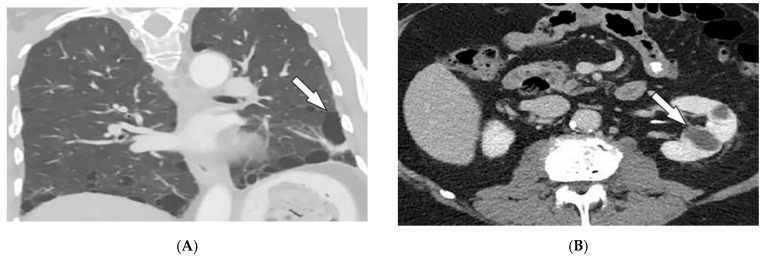
Radiologic images representative of Birt–Hogg–Dube syndrome: (**A**) Coronal CT of the chest in lung windows in a 44-year-old male demonstrates multiple bilateral basilar predominant lentiform cysts abutting the pleura (arrow). (**B**) Axial contrast-enhanced CT of the abdomen shows two left renal cysts (arrow).

**Figure 15 diagnostics-12-02011-f015:**
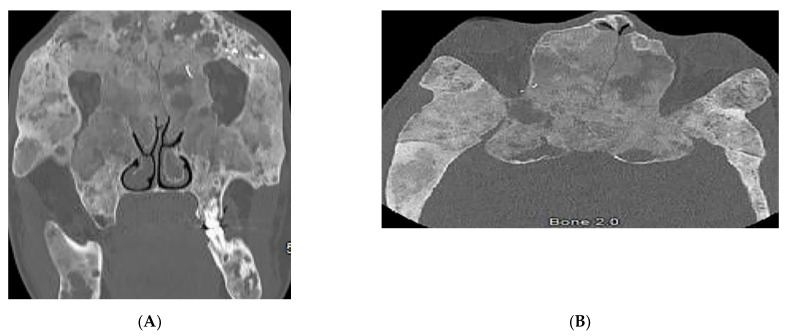
24-year-old female with McCune–Albright syndrome: (**A**) coronal and (**B**) axial maxillofacial CT demonstrate ground-glass expansile appearance of bony structures, a representation of craniofacial fibrous dysplasia.

**Figure 16 diagnostics-12-02011-f016:**
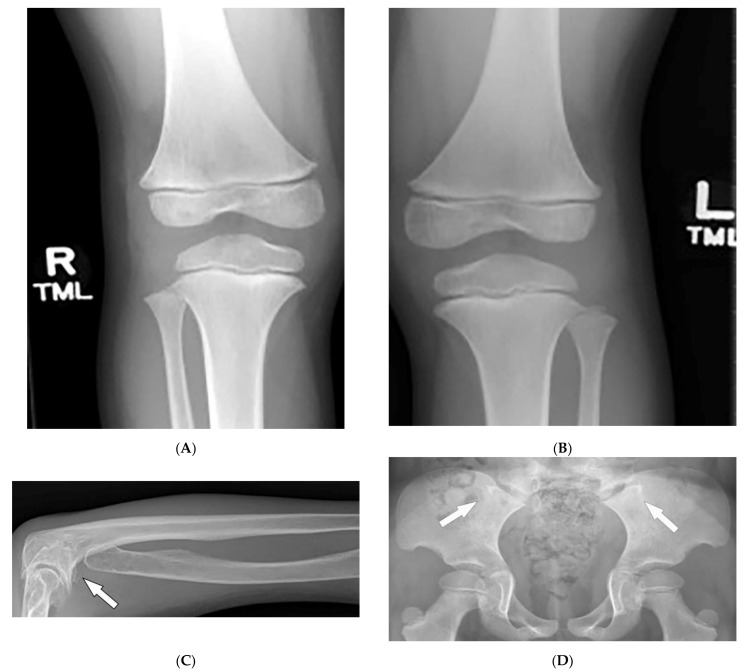
Radiologic images characteristic of Fong (nail–patella) syndrome: (**A**,**B**) AP radiographs of the knees in a 14-year-old male demonstrate a bilateral absence of patellae. (**C**) Radiograph of the forearm in a 56-year-female displays the absence of the radial head (arrow). (**D**) AP radiograph of the pelvis presents bilateral posterior iliac horns (arrows).

**Figure 17 diagnostics-12-02011-f017:**
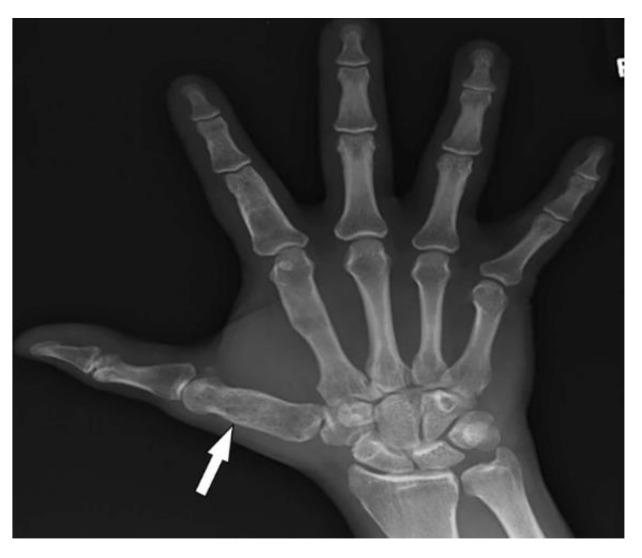
A 30-year-old male with Maffucci syndrome: AP radiograph of the hand demonstrates multiple enchondromas.

**Figure 18 diagnostics-12-02011-f018:**
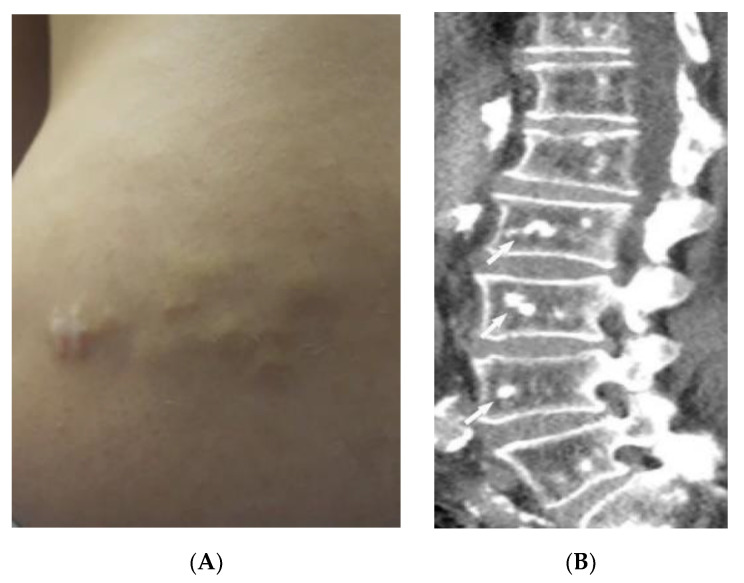
A 60-year-old female with Buschke–Ollendorff syndrome: (**A**) Connective tissue nevi on lower back. (**B**) Sagittal CT of the lumbar spine shows multiple bone islands (arrows). (**C**) AP radiograph of the right knee represents melorheostosis of the tibia (arrow).

**Figure 19 diagnostics-12-02011-f019:**
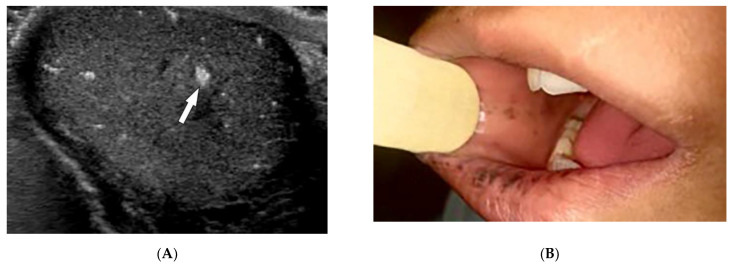
Dermatologic and radiologic images representative of Peutz–Jeghers syndrome: (**A**) Ultrasound of the testis in a 22-year-old male demonstrates testicular lipomatosis (arrow). (**B**) Multiple dark brown lentigines on the mucosal lips and buccal mucosa. (**C**) Coronal MR enterography depicts multiple small-bowel polyps (arrow).

**Figure 20 diagnostics-12-02011-f020:**
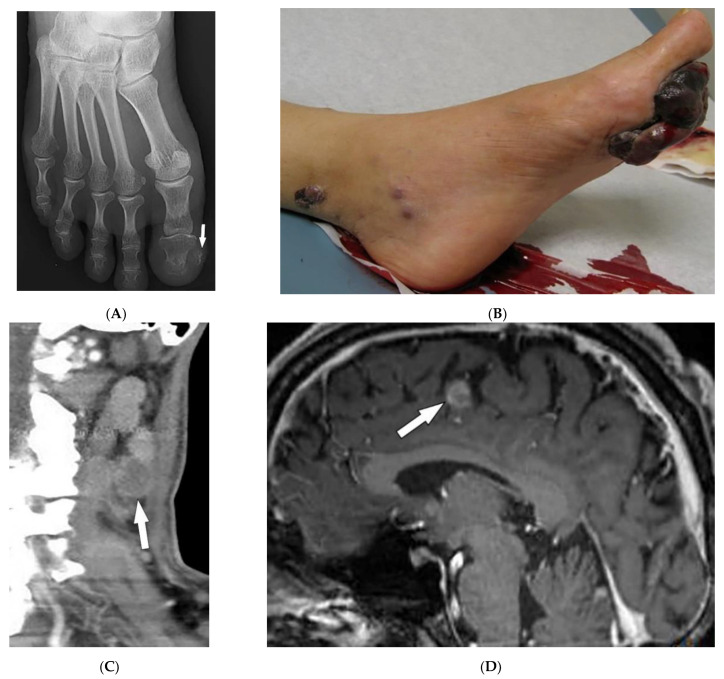
Dermatologic and radiologic images illustrative of melanoma: (**A**) PA radiograph of the foot of a 26-year-old female demonstrated soft-tissue swelling of the great toe with mild calcification medial to the great toe distal phalanx (arrow). (**B**) Large and irregular exophytic plaque on the sole of the left foot. Biopsy confirmed melanoma. (**C**) Sagittal CT of the neck (soft-tissue window) shows a necrotic metastatic lymph node (arrow). (**D**) Sagittal post-contrast T1WI of the brain depicts an enhancing intracranial metastasis (arrow).

**Figure 21 diagnostics-12-02011-f021:**
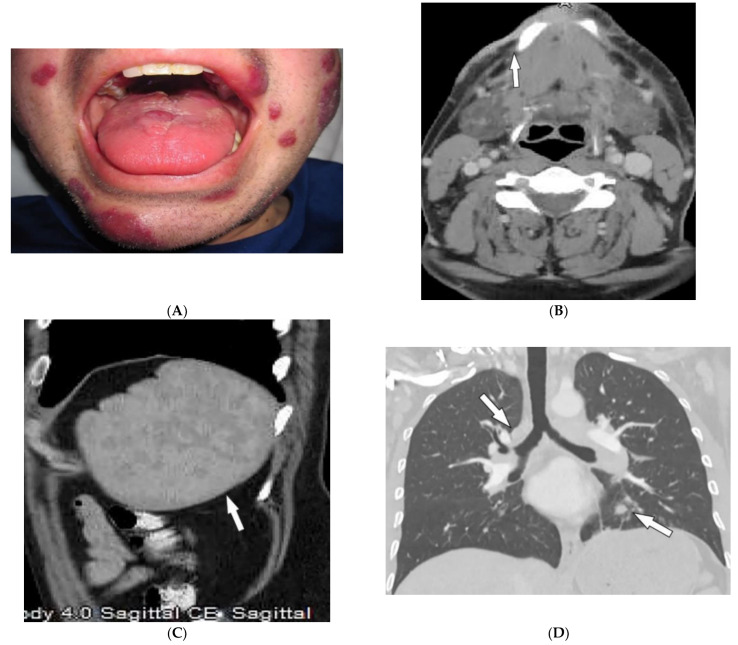
Dermatologic and radiologic images representative of Kaposi sarcoma: (**A**) Red-purple plaques and nodules on the face of a 28-year-old male. (**B**) Axial contrast-enhanced CT of the neck demonstrates diffuse soft-tissue thickening overlying the right mandible (arrow). (**C**) Sagittal contrast-enhanced CT of the abdomen shows splenomegaly (arrow) with heterogeneous enhancement proved to be splenic involvement by Kaposi sarcoma. (**D**) Coronal CT of the chest shows marked peribronchovascular distribution of the tumor with scattered parenchymal nodularity (arrow).

**Table 1 diagnostics-12-02011-t001:** Clinical, Dermatologic, and Imaging Findings in Systemic Diseases.

Disorder	Clinical and Dermatologic Findings	Imaging Findings
**Autoimmune/Inflammatory Disorders and Vasculitides**		
**Dermatomyositis**	Atrophic dermal papules of dermatomyositis (Gottron papules), Gottron sign, heliotrope rash, V sign, shawl sign, calcinosis cutis Proximal nailfold erythema, capillary loop dilation and dropout, ragged cuticles Esophageal dysmotility Myositis ILD Malignancy	Calcinosis cutis Feathery edema-like SI of the muscles NSIP, OP, UIP
**Sarcoidosis**	Lupus pernio Erythema nodosum Lung nodules and adenopathy neurosarcoidosis Bone lesions	Reticulonodular lung opacities with upper lobe and peri-lymphatic distribution Leptomeningeal enhancement Lacy lytic bone lesions
**Scleroderma (diffuse systemic sclerosis)**	Raynaud’s phenomenon Skin tightening Sclerodactyly Calcinosis cutis Dilated bowel/esophagus Pulmonary hypertension ILD	Soft-tissue calcifications and acro-osteolysis Lack of peristalsis and esophageal dilation NSIP and UIP
**Celiac disease**	Dermatitis herpetiformis Psoriasis Intestinal manifestations	Small-bowel dilation Reversal of jejunal and ileal folds
**Granulomatosis with polyangiitis (Wegner’s)**	Palpable purpura Subcutaneous nodules Pyoderma-gangrenosum-like ulcerations Lung lesions and hemoptysis Glomerulonephritis Peripheral neuropathy, mononeuritis multiplex Chronic sinusitis and saddle nose deformity	Bilateral cavitary lung lesions with a ground-glass halo sign Mucosal thickening Nasal septal perforation Hyperostosis
**Polyarteritis nodosa**	Palpable purpura Painful nodules on lower legs Livedo reticularis Medium-sized artery vasculitis	Microaneurysms and constrictions of medium-sized arteritis (beaded appearance)
**Behcet’s disease**	Oral and genital ulcers Ocular findings Vasculitis CNS lesions	Thickening of the aorta and SVC Bilateral pulmonary artery aneurysms Basal ganglia and brainstem lesions
**Genetic/Congenital Disorders**		
**Tuberous sclerosis complex**	Facial angiofibroma Hypopigmented macules Shagreen patches Periungual fibromas Osseous abnormalities CNS hamartomas Renal AML Pulmonary LAM	Tubers, RMLs, SENs, SEGAs of brain Focal sclerotic bone lesions Hypertrophic osteoarthropathy Fat-containing renal mass Thin-walled lung cysts
**Neurofibromatosis type 1 (NF-1)**	Café-au-lait spots Freckling (axillary or inguinal) Lisch nodules Neurofibromas Optic nerve and other gliomas Skeletal abnormalities	Peripheral nerve sheath tumors including cutaneous, spinal, plexiform neuroma Diffuse thickening of the nerve
**Sturge–Weber syndrome**	Port-wine stains Leptomeningeal capillary malformation Glaucoma	Parieto-occipital cortical hemiatrophy Tram-track calcification Calvarial thickening
**PHACES syndrome**	Craniofacial hemangiomas Posterior fossa malformations Cerebrovascular anomalies Eye anomalies	Ipsilateral cerebellar hemisphere dysplasia Major cerebral vessels dysplasia
**Basal cell nevus syndrome**	Basal cell carcinomas Palmoplantar pits Skeletal abnormalities Brain abnormalities	Keratocystic odontogenic tumors Ribs and metacarpals abnormalities Medulloblastoma, falx cerebri calcification
**Hereditary hemorrhagic telangiectasia**	Recurrent epistaxis Multiple telangiectasias Arteriovenous malformations	Bilateral well-defined lung opacities with lobulated shapes Ground-glass nodule
**Birt–Hogg–Dube syndrome**	Fibrofolliculomas, trichodiscomas, acrochordons Lung cysts (pneumothorax) Renal cysts	Bilateral basilar predominant, thin-walled cysts abutting pleura and pulmonary vessels
**McCune–Albright syndrome**	Cafe’-au-lait macules Fibrous dysplasia Endocrine dysfunction	Medullary ground-glass lytic lesions with thin cortices Various sclerotic to cystic pattern
**Fong (Nail–patella) syndrome**	Hypoplastic nails, triangular lunulae Hypoplastic patellae Focal segmental glomerulosclerosisLester iris	Bilateral absence of patellae Posterior iliac horns (Fong’s prongs) Subluxation of radial heads
**Maffucci syndrome**	Multiple enchondromatosis (Ollier disease) Venous malformations	Multiple osteochondromas
**Buschke–Ollendorff syndrome**	Dermatofibrosis lenticularis disseminata Osteopoikilosis Melorheostosis	Bony islands and multiple sclerotic lesions cause mottled appearance Cortical thickening with undulating bone
**Peutz–Jeghers syndrome**	Mucocutaneous pigmented macules Hamartomatous polyps	Multiple intraluminal filling defects on barium study
**Neoplasm**		
**Melanotic melanoma**	ABCDE features	Enhancing lesions if they contain a sufficient amount of melanin Multiple well-defined lung nodules
**Kaposi sarcoma**	Erythematous or violaceous macules, plaques, nodules Pulmonary involvement Gastrointestinal involvement	Nodular enhancing masses Peribroncovascular nodules and halo sign

ILD = interstitial lung disease; SI = signal intensities; NSIP = nonspecific interstitial pneumonia; OP = organizing pneumonia; UIP = usual interstitial pneumonia; CNS = central nervous system; SVC = superior vena cava; RML = radial migration lines; SENs = subependymal nodules; SEGAs = subependymal giant cell astrocytomas; ABCDE = asymmetry, irregular border, color variegation, diameter greater than 6 mm, and evolving morphology.

## Data Availability

This article is a review and not an original study. All the references are listed.

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
