# Peer review of "Imaging More than Skin-Deep: Radiologic and Dermatologic Presentations of Systemic Disorders"

_diagnostics, 2022, doi:10.3390/diagnostics12082011_

Round 1

Reviewer 1 Report

Reviewer comments

Title Imaging More than Skin Deep: Radiologic and Dermatologic Presentations of Systemic Disorders

The manuscript discusses an important point of interest. The manuscript is well organized and well written

Line 40…Table 1. Dermatologic and Imaging Findings in Systemic Disease …diseases not disease

The title of this table is about dermatological and imaging ...However, the table includes the clinical data

Bechet’s ...not correct spelling

English editing is required

The manuscript is illustrated by multiple figures which is very interesting. However, some of which illustrated rare case. The author should determine if these images are theirs or they copy them from other sources

Author Response

The responses, and explanations related to their comments are listed below:

Reviewer 1:

The manuscript discusses an important point of interest. The manuscript is well organized and well written

  • Line 40…Table 1. Dermatologic and Imaging Findings in Systemic Disease …diseases not disease

  • Authors’ response and Action:

Thank you for this comment. We agree with the Reviewer’s comment. We have reviewed our manuscript and changed the word disease to Diseases.

  • The title of this table is about dermatological and imaging ...However, the table includes the clinical data
  • Authors’ response and Action:

We agree with the Reviewer’s comment. We have reviewed our manuscript and changed Dermatologic and Imaging Findings to Clinical, Dermatologic and Imaging Findings.

  • Bechet’s ...not correct spelling
  • Authors’ response and Action:

We agree with the Reviewer’s comment. We have reviewed our manuscript and changed Bechet’s to Behcet’s

  • English editing is required
  • Authors’ response and Action:

We agree with the Reviewer’s comment. We have reviewed our manuscript and improved it significantly.

  • The manuscript is illustrated by multiple figures which is very interesting. However, some of which illustrated rare case. The author should determine if these images are theirs or they copy them from other sources
  • Authors’ response and Action:

We have the copyright of all included images.

Reviewer 2 Report

1) Abstract L14-18. Cutaneous manifestations of systemic diseases are diverse and sometimes precede more serious disease and symptomatology. Similarly, radiologic imaging plays a key role in early diag-  nosis and determination of extent of systemic involvement. To improve diagnostic accuracy and patient care, it is important that clinicians and radiologists be familiar with both cutaneous and  radiologic features of various systemic disorders. This article reviews cutaneous manifestations and  imaging findings of commonly encountered systemic diseases. Could you please add a section on background?

 2) 1) Abstract L14-18. Cutaneous manifestations of systemic diseases are diverse and sometimes precede more serious disease and symptomatology. Similarly, radiologic imaging plays a key role in early diag-  nosis and determination of extent of systemic involvement. To improve diagnostic accuracy and patient care, it is important that clinicians and radiologists be familiar with both cutaneous and  radiologic features of various systemic disorders. This article reviews cutaneous manifestations and  imaging findings of commonly encountered systemic diseases. The abstract is quite rumbling could you please divide it in different sections (i.e. background, aims, conclusions, ..) 

3) Table 1. Dermatologic and Imaging Findings in Systemic Disease. Could you please add for scleroderma the imaging evaluation of skin thickness by ultrasound and elastography and skin perfusion by laser techniques  as reported in several articles:

A) Correlations between blood perfusion and dermal thickness in different skin areas of systemic sclerosis patients. Microvasc Res. 2018 Jan;115:28-33. doi: 10.1016/j.mvr.2017.08.004. Epub 2017 Aug 20. PMID: 28834709.

B) Evaluation of blood perfusion by laser speckle contrast analysis in different areas of hands and face in patients with systemic sclerosis. Ann Rheum Dis. 2014 Nov;73(11):2059-61. doi: 10.1136/annrheumdis-2014-205528. 

c) High-resolution ultrasound imaging of skin involvement in systemic sclerosis: a systematic review. Rheumatol Int. 2021 Feb;41(2):285-295. doi: 10.1007/s00296-020-04761-8. 

3) page 5 of 15. Please correct this typo: HRTC instead of CT for the high-resolution computed tomography

4) 2.2. Sarcoidosis L99-102. Sarcoidosis is a granulomatous disease with unknown etiology and variable prevalence, with a predilection for African American women in their 3rd to 5th decades of life  [12-15]. Patients may be asymptomatic or experience a wide spectrum of multi-organ involvement [13, 14] demanding radiologic investigation for diagnosis and follow-up [12, 103 13, 15]. Plese improve this paragraph and add these references:

A- Sarcoidosis: A Clinical Overview from Symptoms to Diagnosis. Cells. 2021 Mar 31;10(4):766. doi: 10.3390/cells10040766.

B - Correlation between Potential Risk Factors and Pulmonary Embolism in Sarcoidosis Patients Timely Treated. J Clin Med. 2021 Jun 2;10(11):2462. doi: 10.3390/jcm10112462. 

5) 2.3 Scleroderma. L 165-169. The skin is the main  organ involved in scleroderma, and disease subsets are differentiated by degree of skin  involvement [21]. Raynaud’s phenomenon, cutaneous sclerosis, nailfold and fingernail al terations, cutaneous ulcerations, telangiectasias, “salt and pepper” hyper/hypopigmenta tion (Fig. 3B) and calcinosis cutis are common skin manifestations seen in scleroderma patients [4, 21]. Please improve this paragraph and add all the references reported to ameliorate Table 1 (Point 3):

A) Correlations between blood perfusion and dermal thickness in different skin areas of systemic sclerosis patients. Microvasc Res. 2018 Jan;115:28-33. doi: 10.1016/j.mvr.2017.08.004. Epub 2017 Aug 20. PMID: 28834709.

B) Evaluation of blood perfusion by laser speckle contrast analysis in different areas of hands and face in patients with systemic sclerosis. Ann Rheum Dis. 2014 Nov;73(11):2059-61. doi: 10.1136/annrheumdis-2014-205528. 

c) High-resolution ultrasound imaging of skin involvement in systemic sclerosis: a systematic review. Rheumatol Int. 2021 Feb;41(2):285-295. doi: 10.1007/s00296-020-04761-8. 

 6) 2.3 Scleroderma. L 170-171. Cutaneous sclerosis begins in the fingers, extends proximally to metacar pophalangeal joints, and affects the face at an early stage. The skin becomes pale and hairless with the skin folds disappear. Please add ameliorate thia paragraph as reported in these sentences: "Routine clinical practice classifies the skin manifestation of this disorder into three different subsets, limited cutaneous skin involvement (lcSSc), diffuse cutaneous skin involvement (dcSSc) and limited SSc (lSSc). The skin manifestation may be recognized and studied by the modified Rodnan skin score (mRss), the validated method to evaluate the severity of skin involvement in SSc and to distinguish, as aforementioned, patients with lcSSc from those with dcSSc or with lSSc". Please add this reference: The role of ultrasound in systemic sclerosis: On the cutting edge to foster clinical and research advancement. J Scleroderma Relat Disord. 2021 Jun;6(2):123-132. doi: 10.1177/2397198320970394. 

7) L 357-359. Renal angiomyolipomas (AMLs), composed of varying amounts of blood vessels, smooth muscle, and fat, occur in 55–75% of TSC patients, and are commonly multiple and  bilateral. On ultrasonography, AMLs are homogeneous or heterogeneous hyperechoic lesions. Please divide this paragraph by precedent with a subtitle.

8) L 601-607. OPK are asymptomatic dense ‘bony islands’ (Fig. 18B) presenting as numerous well-  defined symmetric densities. These lesions are found incidentally on radiographs as scle rotic densities and give the bone a mottled appearance. In some cases, OPK lesions resem ble osteoblastic metastases. However, normal bone scintigraphy in OPK excludes other differential diagnoses [15, 66-68]. Melorheostosis (Fig. 18C), another BOS association, is a dense, irregular, eccentric hyperostosis of the cortex with a distinct demarcation border that causes irregular thickening of cortical bone with a melting wax appearance on imaging [66, 69]. Please explain all the synonyms (i.e. OPK, BOS, AD, etc ..)  and add this paper:

Coexistence of osteopoikilosis with seronegative spondyloarthritis and Raynaud's phenomenon: first case report with evaluation of the nailfold capillary bed and literature review. Reumatismo. 2012 Dec 11;64(5):335-9. doi: 10.4081/reumatismo.2012.335.

9) 5. Conclusion.

Please improve the conclusions: underline the novelty of the study and ameliorate the description of clinical implications.

Author Response

The responses, and explanations related to their comments are listed below:

Reviewer 2:

  • Abstract L14-18. Cutaneous manifestations of systemic diseases are diverse and sometimes precede more serious disease and symptomatology. Similarly, radiologic imaging plays a key role in early diag- nosis and determination of extent of systemic involvement. To improve diagnostic accuracy and patient care, it is important that clinicians and radiologists be familiar with both cutaneous and  radiologic features of various systemic disorders. This article reviews cutaneous manifestations and  imaging findings of commonly encountered systemic diseases. Could you please add a section on background?
  • Authors’ response and Action:

We agree with the Reviewer’s comment. We have reviewed our manuscript and background is added to the abstract.

  • Abstract L14-18. Cutaneous manifestations of systemic diseases are diverse and sometimes precede more serious disease and symptomatology. Similarly, radiologic imaging plays a key role in early diag- nosis and determination of extent of systemic involvement. To improve diagnostic accuracy and patient care, it is important that clinicians and radiologists be familiar with both cutaneous and  radiologic features of various systemic disorders. This article reviews cutaneous manifestations and  imaging findings of commonly encountered systemic diseases. The abstract is quite rumbling could you please divide it in different sections (i.e. background, aims, conclusions, ..)
  • Authors’ response and Action:

We agree with the Reviewer’s comment. We have reviewed our manuscript and revised the abstract based on your comment. I believe it has improved significantly.

‘Abstract:

Background: Cutaneous manifestations of systemic diseases are diverse and sometimes precede more serious diseases and symptomatology. Similarly, radiologic imaging plays a key role in ear-ly diagnosis and determination of extent of systemic involvement. Simultaneous awareness of skin and imaging manifestations can help the radiologist to narrow differential diagnosis even if imaging findings are nonspecific.

 Aims: To improve diagnostic accuracy and patient care, it is important that clinicians and radiol-ogists be familiar with both cutaneous and radiologic features of various systemic disorders. This article reviews cutaneous manifestations and imaging findings of commonly encountered sys-temic diseases.

Conclusions: Familiarity with the most disease-specific skin lesions help the radiologist pinpoint a specific diagnosis and consequently, preventing unnecessary invasive workups and contrib-uting to improved patient care.’

  • Table 1. Dermatologic and Imaging Findings in Systemic Disease. Could you please add for scleroderma the imaging evaluation of skin thickness by ultrasound and elastography and skin perfusion by laser techniques  as reported in several articles:

  1. Correlations between blood perfusion and dermal thickness in different skin areas of systemic sclerosis patients. Microvasc Res. 2018 Jan;115:28-33. doi: 10.1016/j.mvr.2017.08.004. Epub 2017 Aug 20. PMID: 28834709.

  1. Evaluation of blood perfusion by laser speckle contrast analysis in different areas of hands and face in patients with systemic sclerosis. Ann Rheum Dis. 2014 Nov;73(11):2059-61. doi: 10.1136/annrheumdis-2014-205528.

  1. High-resolution ultrasound imaging of skin involvement in systemic sclerosis: a systematic review. Rheumatol Int. 2021 Feb;41(2):285-295. doi: 10.1007/s00296-020-04761-8.
  • Authors’ response and Action:

We agree with the Reviewer’s comment. ‘Current literature supports using high-frequency ultrasound for quantitative and reliable evaluation of dermal thickness in patients with SSc. Dermal thickness has been shown to be inversely corelated with blood perfusion. Also, US elastography has been shown to be of value in the evaluation of the skin in SSc ’ was added for the application of high-frequency ultrasound. I did not add anything related to laser techniques as it does not fit the scope of our article.

  • page 5 of 15. Please correct this typo: HRTC instead of CT for the high-resolution computed tomography
  • Authors’ response and Action:

We agree with the Reviewer’s comment. We have reviewed our manuscript and changed CT to HRCT wherever appropriate.

  • 2. Sarcoidosis L99-102. Sarcoidosis is a granulomatous disease with unknown etiology and variable prevalence, with a predilection for African American women in their 3rd to 5th decades of life [12-15]. Patients may be asymptomatic or experience a wide spectrum of multi-organ involvement [13, 14] demanding radiologic investigation for diagnosis and follow-up [12, 103 13, 15]. Plese improve this paragraph and add these references:

  • Sarcoidosis: A Clinical Overview from Symptoms to Diagnosis. Cells. 2021 Mar 31;10(4):766. doi: 10.3390/cells10040766.

  • Correlation between Potential Risk Factors and Pulmonary Embolism in Sarcoidosis Patients Timely Treated. J Clin Med. 2021 Jun 2;10(11):2462. doi: 10.3390/jcm10112462.

  • Authors’ response and Action:

We agree with the Reviewer’s comment. The recommended references are added. ‘Parenchymal involvement and pulmonary embolism are other chest manifestations of sarcoidosis.’ Added to sarcoidosis section.

  • 3 Scleroderma. L 165-169. The skin is the main organ involved in scleroderma, and disease subsets are differentiated by degree of skin  involvement [21]. Raynaud’s phenomenon, cutaneous sclerosis, nailfold and fingernail al terations, cutaneous ulcerations, telangiectasias, “salt and pepper” hyper/hypopigmenta tion (Fig. 3B) and calcinosis cutis are common skin manifestations seen in scleroderma patients [4, 21]. Please improve this paragraph and add all the references reported to ameliorate Table 1 (Point 3):

  • Correlations between blood perfusion and dermal thickness in different skin areas of systemic sclerosis patients. Microvasc Res. 2018 Jan;115:28-33. doi: 10.1016/j.mvr.2017.08.004. Epub 2017 Aug 20. PMID: 28834709.

  • Evaluation of blood perfusion by laser speckle contrast analysis in different areas of hands and face in patients with systemic sclerosis. Ann Rheum Dis. 2014 Nov;73(11):2059-61. doi: 10.1136/annrheumdis-2014-205528.

  • High-resolution ultrasound imaging of skin involvement in systemic sclerosis: a systematic review. Rheumatol Int. 2021 Feb;41(2):285-295. doi: 10.1007/s00296-020-04761-8.
  • Authors’ response and Action:

We agree with the Reviewer’s comment. The recommended references are added. ‘Current literature supports using high-frequency ultrasound for quantitative and reliable evaluation of dermal thickness in patients with SSc. Dermal thickness has been shown to be inversely corelated with blood perfusion. Also, US elastography has been shown to be of value in the evaluation of the skin in SSc.’ was added for the application of high-frequency ultrasound. I did not anything related to laser techniques as it does not fit the scope of our article.

  • 3 Scleroderma. L 170-171. Cutaneous sclerosis begins in the fingers, extends proximally to metacar pophalangeal joints, and affects the face at an early stage. The skin becomes pale and hairless with the skin folds disappear. Please add ameliorate thia paragraph as reported in these sentences: "Routine clinical practice classifies the skin manifestation of this disorder into three different subsets, limited cutaneous skin involvement (lcSSc), diffuse cutaneous skin involvement (dcSSc) and limited SSc (lSSc). The skin manifestation may be recognized and studied by the modified Rodnan skin score (mRss), the validated method to evaluate the severity of skin involvement in SSc and to distinguish, as aforementioned, patients with lcSSc from those with dcSSc or with lSSc". Please add this reference: The role of ultrasound in systemic sclerosis: On the cutting edge to foster clinical and research advancement. J Scleroderma Relat Disord. 2021 Jun;6(2):123-132. doi: 10.1177/2397198320970394.
  • Authors’ response and Action:

We agree with the Reviewer’s comment. This sentence and related reference was added too: ‘US elastography has been shown to be of value in the evaluation of the skin in SSc.’

  • L 357-359. Renal angiomyolipomas (AMLs), composed of varying amounts of blood vessels, smooth muscle, and fat, occur in 55–75% of TSC patients, and are commonly multiple and On ultrasonography, AMLs are homogeneous or heterogeneous hyperechoic lesions. Please divide this paragraph by precedent with a subtitle.
  • Authors’ response and Action:

We agree with the Reviewer’s comment. This part changed to:

‘Renal involvement includes angiomyolipomas (AMLs), which are composed of varying amounts of blood vessels, smooth muscle, and fat, occur in 55–75% of TSC pa-tients, and are commonly multiple and bilateral. On ultrasonography, AMLs are homogeneous or heterogeneous hyperechoic lesions.’

L 601-607. OPK are asymptomatic dense ‘bony islands’ (Fig. 18B) presenting as numerous well-  defined symmetric densities. These lesions are found incidentally on radiographs as scle rotic densities and give the bone a mottled appearance. In some cases, OPK lesions resem ble osteoblastic metastases. However, normal bone scintigraphy in OPK excludes other differential diagnoses [15, 66-68]. Melorheostosis (Fig. 18C), another BOS association, is a dense, irregular, eccentric hyperostosis of the cortex with a distinct demarcation border that causes irregular thickening of cortical bone with a melting wax appearance on imaging [66, 69]. Please explain all the synonyms (i.e. OPK, BOS, AD, etc ..)  and add this paper:

Coexistence of osteopoikilosis with seronegative spondyloarthritis and Raynaud's phenomenon: first case report with evaluation of the nailfold capillary bed and literature review. Reumatismo. 2012 Dec 11;64(5):335-9. doi: 10.4081/reumatismo.2012.335.

  • Authors’ response and Action:

We agree with the Reviewer’s comment. All abbreviations are added in extended form. The reference was also added.

9) 5. Conclusion.

Please improve the conclusions: underline the novelty of the study and ameliorate the description of clinical implications.

  • Authors’ response and Action:

We agree with the Reviewer’s comment. We have reviewed our manuscript and revised the conclusion based on your comment. I believe it has improved significantly. ‘Conclusion

Various systemic conditions have specific or non-specific dermatologic and imag-ing features. Simultaneous consideration of imaging findings and dermatologic mani-festations helps in more precise imaging interpretation and narrows the differential diagnosis toward the final diagnosis. Sometimes the cutaneous manifestation of a sys-temic disease is predictive of systemic involvement e.g., pulmonary hypertension and ILD in lcSSc vs. dSSc, respectively.  Familiarity with the most disease-specific skin le-sions help the radiologist pinpoint a specific diagnosis and consequently, preventing unnecessary invasive workups and contributing to improved patient care.All abbreviations are added in extended form. The reference was also added.’